# Structural phylogenetics unravels the evolutionary diversification of communication systems in gram-positive bacteria and their viruses

**David Moi** [1,2,8] ✉, **Charles Bernard**[1,2,8], **Martin Steinegger**[3,4,5], **Yannis Nevers**[1,2], **Mauricio Langleib**[6,7] **& Christophe Dessimoz** [1,2] ✉

Recent advances in artificial-intelligence-based protein structure modeling have yielded remarkable progress in predicting protein structures. Because structures are constrained by their biological function, their geometry tends to evolve more slowly than the underlying amino acids sequences. This feature of structures could in principle be used to reconstruct phylogenetic trees over longer evolutionary timescales than sequence-based approaches; however, until now, a reliable structure-based tree-building method has been elusive. Here, we introduce a rigorous framework for empirical tree accuracy evaluation and tested multiple approaches using sequence and structure information. The best results were obtained by inferring trees from sequences aligned using a local structural alphabet—an approach robust to conformational changes that confound traditional structural distance measures. We illustrate the power of structure-informed phylogenetics by deciphering the evolutionary diversification of a particularly challenging family: the fast-evolving RRNPPA quorum-sensing receptors. We were able to propose a more parsimonious evolutionary history for this critical protein family that enables gram-positive bacteria, plasmids and bacteriophages to communicate and coordinate key behaviors. The advent of high-accuracy structural phylogenetics enables a myriad of applications across biology, such as uncovering deeper evolutionary relationships, elucidating unknown protein functions or refining the design of bioengineered molecules.

Since Darwin, phylogenetic trees have depicted evolutionary relationships among organisms, viruses, genes and other evolving entities, allowing us to understand their shared ancestry and tracing the events that led to the observable extant diversity. Trees based on molecular data are typically reconstructed from nucleotide or amino acid sequences by aligning homologous sequences and inferring the tree topology and branch lengths under a model of character substitution[1–3]. However, over long evolutionary timescales, multiple substitutions occurring at the same site cause uncertainty in alignment and tree building. The problem is particularly acute when dealing with fast-evolving sequences, such as viral or immune-related ones, or when attempting to resolve distant relationships, at the origin of animals[4–6] or beyond.

In contrast, the fold of proteins is often conserved well past sequence signal saturation. Furthermore, because three-dimensional (3D) structure determines function, protein structures have long

**Fig. 1 | Benchmarking results for alignment and tree building methods.**
**a**, A diverse set of tree-building methods using amino acid and/or structure information were tested (Methods and Supplementary Information). **b**, Among the approaches tested, the one incorporating structure information in the alignment phase (FoldTree) exhibited the highest proportion of trees with the highest TCS on the OMA dataset (protein families defined from sequences; relatively close with $n = 4,592$ families). By contrast, structure trees from LDDT and TM underperformed compared to sequence trees and maximum-likelihood trees built with partition models of sequence and structural characters. Error bars are derived from the variance of a multinomial distribution with the identical class probabilities and $n = 4,592$ draws. **c**, In the CATH dataset of structurally defined protein families ($n = 488$ families), FoldTree metrics garner a higher proportion of trees with the highest TCS per family. The proportion of highest-scoring trees using structurally informed methods is greater than in the families defined in the OMA dataset. Error bars are derived from the variance of a multinomial distribution with the empirically derived class probabilities and $n = 488$ draws. **d**, The variance of normalized root-to-tip distances was compiled for all trees within the OMA dataset for all structural tree methods and sequence trees. FoldTree has a lower mean variance than other methods. The mean of each distribution is shown with a vertical red line. Distributions are truncated to values between 0 and 0.2. **e**, A random sample of trees is shown where each column is from from equivalent protein input sets and each column of trees is derived using a distinct tree-building method.

been studied to gain insight into their biological role within the cell, whether it be catalyzing reactions, interacting with other proteins to form complexes or regulating the expression of genes, among others. Until recently, protein structures had to be obtained through labor-intensive crystallography and other experimental methods, with modeling efforts often falling short of the level of accuracy required for the many tasks structures were used for. Because of these limitations, structural biology and phylogenetics have developed

as largely separate disciplines and each field has created different models describing evolutionary or molecular phenomena suited to the availability of computational power and experimental data. Despite these limitations, attempts have been made to merge the two paradigms[7–9].

Now, the widespread availability of accurate structural models based on artificial intelligence (AI) predictions[10,11] opens up the prospect of reconstructing trees from structures. However, there are pitfalls to avoid to derive evolutionary distances between homologous protein structures. Geometric distances between rigid-body representations of structures, such as root-mean-square deviation (r.m.s.d.) distance or template modeling (TM) score[12], are confounded by spatial variations caused by conformational changes[13,14]. More local structural similarity measures have been proposed in the context of protein classification[13]; but because of the relative paucity of available structures until recently, little is known about the accuracy of structure-based phylogenetic reconstruction beyond a few isolated case studies[15,16].

Here, we report the large-scale comprehensive evaluation of phylogenetic trees reconstructed from the structures of thousands of protein families across the tree of life, using multiple kinds of distance measures and tree-building strategies. We tested nine structure-informed approaches, using divergence measures obtained using Foldseek[17], which outputs scores from rigid-body alignment, local superposition-free alignment and structural alphabet-based sequence alignments. In addition, we tested a recently proposed partitioned structure and sequence likelihood method[18]. The performance of these approaches has been previously assessed on the task of detecting whether folds are homologous and belong to the same family[17,19,20] or on a few examples[18] but have not been systematically evaluated for phylogenetic tree inference. Remarkably, we found that some but not all structure-informed approaches are competitive with state-of-the-art sequence-based phylogenetic methods and outperform them on highly divergent datasets across benchmarks related to tree topology (species tree branch support based on taxonomic congruence score (TCS) and accurate species tree algorithm (ASTRAL)) and when testing the adherence to a molecular clock.

To demonstrate the capabilities of structural phylogenetics, we use the current best approach (which we have named FoldTree), released as open-source software, to resolve the difficult phylogeny of a fast-evolving protein family of high relevance: the RRNPPA (Rap, Rgg, NprR, PlcR, PrgX and AimR) receptors of communication peptides. Although these receptors were identified in the early 1990s[21,22], their evolutionary history is unclear because of frequent mutations, making sequence comparisons challenging[23–25]. This is reflected by the nomenclature of the family, whereby these proteins were historically described as six different families of intracellular receptors and only structural comparisons allowed to establish the actual consensus on their common evolutionary origin[25–27]. These proteins allow gram-positive bacteria, their plasmids and their viruses to assess their population density and regulate key biological processes accordingly. These communication systems have been shown to regulate virulence, biofilm formation, sporulation, competence, solventogenesis, antibiotic resistance or antimicrobial production in bacteria[28–32], conjugation in conjugative elements, lysis-lysogeny decision in bacteriophages[33] or host manipulation by mobile genetic elements[30,34]. These receptors are paired with a small secreted communication peptide that accumulates extracellularly as the encoding population replicates. Once a quorum of cells, plasmids or viruses is met, communication peptides get frequently internalized within cells and binds to the tetratricopeptide repeats (TPRs) of cognate intracellular receptors, leading to gene or protein activation or inhibition and facilitating a coordinated response beneficial for a dense population. Notably, the RRNPPA family impacts human health as it connects to the virulence and transmissibility of pathogenic bacteria, and to the spread of antimicrobial resistance genes through regulation of

horizontal gene transfers. We analyze and discuss the relative parsimony of the phylogeny of this family, highlighting the contrasts with the sequence-based tree.

## Results

### Structure-informed trees can outperform sequence-only trees

To incorporate structural information in phylogenetic tree building, we investigated the use of local superposition-free comparison (local distance difference test; LDDT[20]), rigid-body alignment (TM score[12]) and a distance derived from a statistically corrected sequence similarity after aligning with a structural alphabet (Fident[17]). These measures were used to compute distance trees using neighbor joining (NJ) after being aligned in an all-versus-all comparison using the Foldseek structural alphabet (Methods). The results of our empirical benchmarking strategy are summarized in Fig. 1. The top performing tree building pipeline of the assessment was then used to derive the RRNPPA phylogeny shown in Fig. 2.

Assessing the accuracy of trees reconstructed from empirical data is notoriously difficult. We used two complementary indicators, 'correct' topology and adherence to a molecular clock. We designed the Taxonomic Congruence Score (TCS; (Supplementary Notes and Fig. 1) to assess the congruence of reconstructed protein trees with the known taxonomy[35]. Among several potential tree topologies reconstructed from the same set of input proteins, the 'better' topologies can be expected to have a higher TCS on average. This is because of the fact that, in the majority of cases, gene families are inherited vertically in cellular organisms from parent to daughter cells[36]. The TCS metric used in these benchmarks is designed to weigh topological congruence closer to the root more heavily than toward the leaves.

For trees reconstructed from closely related protein families using standard sequence alignments (the 'OMA dataset'; Methods), we thoroughly tested nine approaches using combinations of structure and sequence input data paired with either multiple-sequence alignment (MSA) and maximum-likelihood approaches to build trees or distance-based NJ trees (Figs. 2a and 3). Trees derived from the combination of sequence and structural alignment based on the statistically corrected Fident distance (henceforth referred to as the FoldTree approach) outperformed other approaches in garnering the highest percentage of top scoring trees within this dataset (Fig. 2b). This trend was observed across various protein family subsets, taken from taxonomically defined clades within the species tree at varying evolutionary distances (Extended Data Fig. 1). We also experimented with other parameter variations but they did not lead to further improvements (Extended Data Figs. 1–3). In particular, we did not see an improvement when we used only the 3Di structural alphabet in both the alignment and the distance estimation steps using the distance and maximum-likelihood tree-building strategies, indicating that our use of structures mainly contributes to better identifying the homologous residues (Extended Data Fig. 2).

We then compared FoldTree with other methods over larger evolutionary distances, using structure-informed homologous families from the CATH database[37]. This database classifies proteins hierarchically, grouping them on the basis of class, architecture, topology and homology of experimentally determined protein structures. We examined sets of proteins from the same homology set after filtering for redundancy (Methods). Efforts were made to correct crystal structures with discontinuities or other defects before tree building (Methods) as these adversely affect structural comparisons. When measuring TCS in families from this more divergent CATH dataset, structure-based methods performed better overall. FoldTree outperformed the sequence-based methods by a larger margin and structurally informed trees received a larger proportion of highest-scoring trees overall (Fig. 2c). In particular, structurally informed maximum-likelihood trees[18] also benefited relative to purely sequence-based methods. Nevertheless, in a pairwise comparison between this method and FoldTree, the simpler

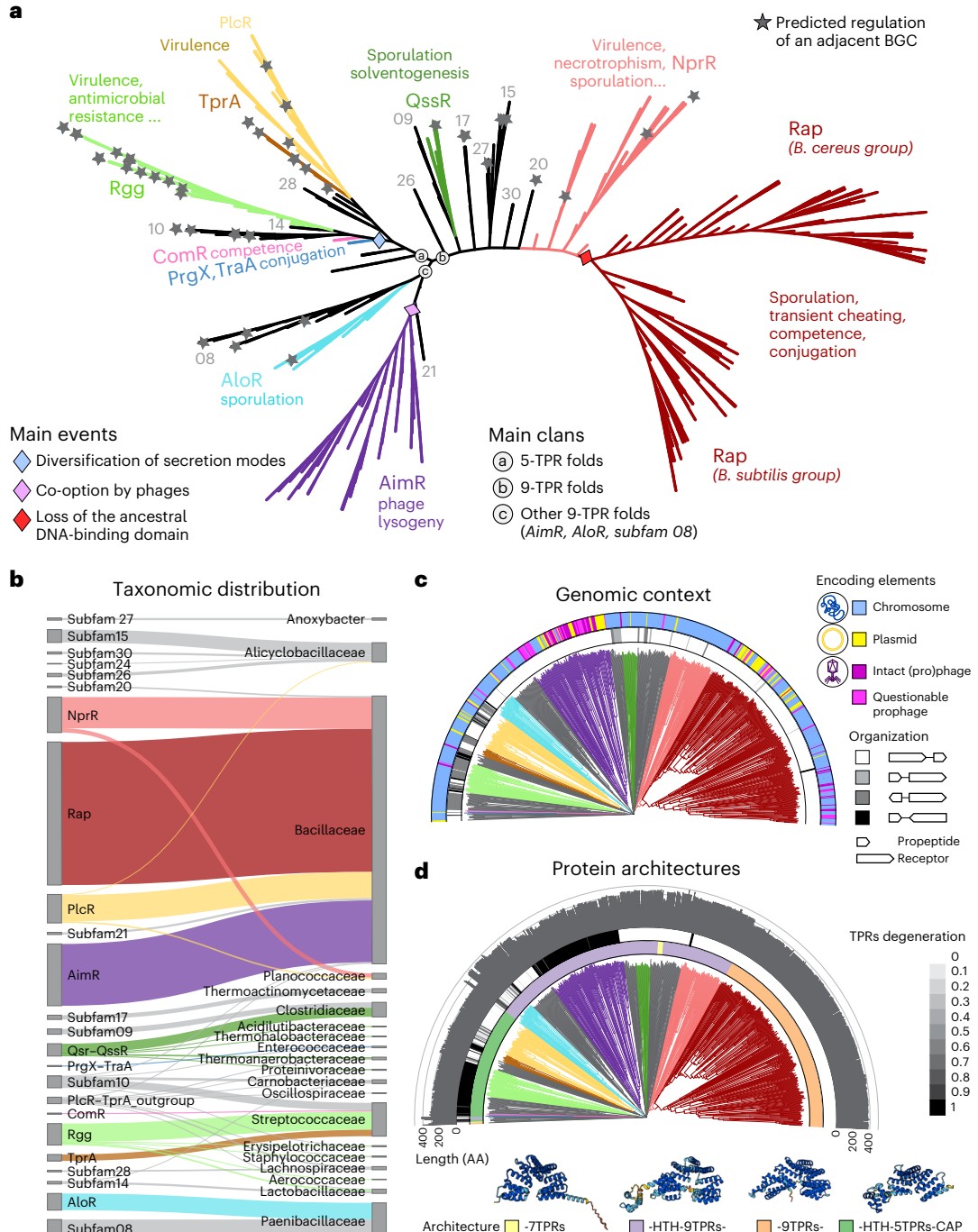

**Fig. 2 | Phylogeny of cytosolic receptors from the RRNPPA family paired with a communication propeptide. a**, Functional diversity of the RRNPPA family. The tree is unrooted and branch lengths are ignored for visualization purposes (displayed in **c,d**). Subfamilies with experimental validation of at least one member are highlighted in color. The other subfamilies depicted with a number in gray correspond to high-confidence candidate subfamilies detected with RRNPP_detector[30]. Biological processes experimentally shown to be regulated in a density-dependent manner by a quorum-sensing system are displayed for each validated subfamily. A star mapped to a leaf indicates a predicted regulation of an adjacent biosynthetic gene cluster by the corresponding quorum-sensing system. **b**, Distribution and prevalence of the different members of each RRNPPA subfamily into the different taxonomic families. **c**, Genomic orientation and encoding element of the receptor and adjacent propeptide pairs. **d**, The first color strip indicates the domain architecture of each receptor. A representative fold for each domain architecture is displayed in the legend (AlphaFold models of subfamily 27, NprR, Rap and PlcR, respectively). The second color strip gives the degeneration score of TPR sequences of each receptor (given as 1 − TprPred_likelihood[26]). The histogram shows the length (in amino acids) of each receptor.

evolutionary model of FoldTree outperformed this approach in terms of TCS (Extended Data Fig. 4). Additional pairwise comparisons between structural distance methods and maximum-likelihood trees showed that structural methods provide more congruent topologies with CATH protein families that allow for greater evolutionary distances (Extended Data Fig. 7).

Using both the CATH and the OMA datasets, we performed an additional benchmark, recently proposed[38] as a reaction to an earlier version of this work. This benchmark is conceptually similar to TCS but puts more weight on topological differences to the reference species tree close to the leaves (Supplementary Notes). This benchmark yielded generally consistent results (Extended Data Figs. 8 and 9) but on the

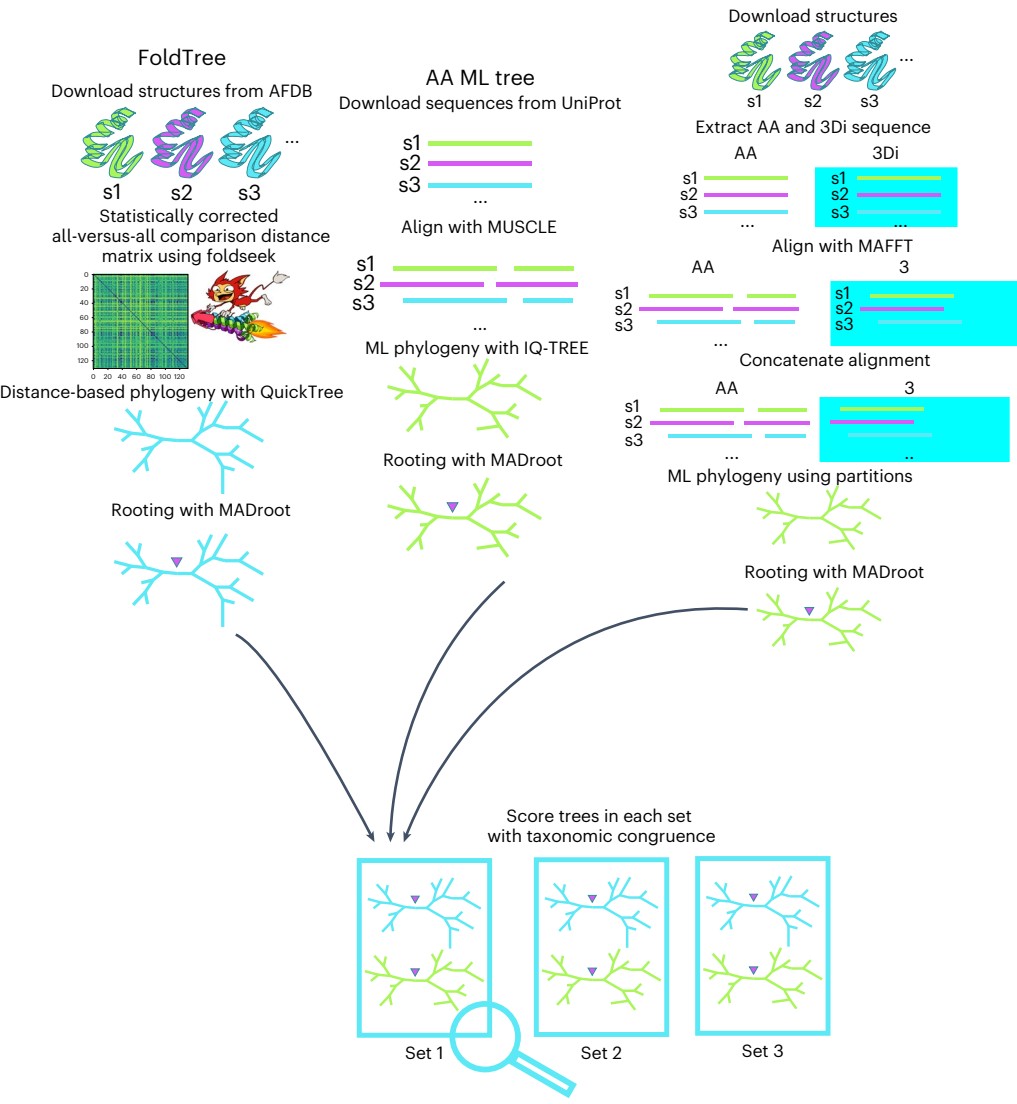

**Fig. 3 | Benchmarking pipeline schematic of trees created from equivalent protein sets for structure and sequence trees.** On the structural side of the pipeline, all-versus-all comparisons using Foldseek are compiled into a statistically corrected distance matrix. These are used as input for QuickTree and rooted with MAD. On the sequence side of the pipeline, the sequences are aligned with MUSCLE. A maximum-likelihood tree is derived using IQ-TREE and then rooted with MAD. The method described by Puente-Lelievre et al. is implemented as reported in their manuscript[18]. The 3Di and amino acid characters are aligned using MAFFT and then concatenated into an alignment containing both with the same pattern of gaps in both partitions. IQ-TREE is then used with a partition model using the LG and 3Di substitution matrices on their respective partitions. Both structural and sequence-based trees are scored using the TCS metric described in the Supplementary Notes and outlined in Fig. 2.

closer families defined by OMA, FoldTree is slightly outperformed by a sequence-based maximum-likelihood approach[38]. However, in more divergent families defined by CATH, FoldTree retains an advantage over that approach (Extended Data Fig. 9).

We found that filtering input families on the basis of the confidence of the AlphaFold structure prediction (predicted local distance difference test LDDT (pLDDT)) increased the proportion of trees where FoldTree was outperforming maximum-likelihood models (Extended Data Fig. 3). This suggests that advancements in structural prediction might further benefit structural trees in the future. We also characterized the distributions of pairwise identities within the induced pairs of sequence-based and 3Di-based MSAs, as well as Foldseek-based MSAs, which combine both alphabets. Here, we found that the distribution of mean pairwise percentage identity within alignments was increased by the use of Foldseek and leveraging structure and sequence in tandem to align residues (Extended Data Fig. 1). We also investigated the relationship between sequence divergence within families and the observed differences in TCS between FoldTree and maximum-likelihood approaches. We found that, as the mean amino acid percentage identity decreased, the proportion of FoldTree trees with better topologies than the maximum-likelihood trees increased (Extended Data Fig. 6).

To validate our findings using another independent indicator of tree quality, we also assessed how uniform a tree's root-to-tip lengths are for all its tips, akin to following a molecular clock. Although strict adherence to a molecular clock is unlikely in general, it is reasonable to assume that distance measures resulting in more monotonic trees on average (that is, with reduced root-to-tip variance; Methods) are more accurate[39]. However, this in and of itself is not a strong indicator of tree quality and must be considered in conjunction with correct topology using a metric such as TCS or ASTRAL quartet support values. We found that, in the OMA sequence-based family dataset, FoldTree had the lowest root-to-tip variance of all approaches (Fig. 2d).

**Fig. 4 | Corecut pipeline schematic.** In steps 1–5, we outline how we used Foldseek to derive a consensus core structure between a group of homologs. This process divides the structure into three regions. In the second part of the pipeline (steps 6–10) we show how we constructed clusters for the N and C terminals and a FoldTree from the consensus core of the set of homologs. The final output of this pipeline is a FoldTree of the core labeled with the assigned clusters of the C and N termini of the input structures. AFDB, AlphaFold Structure Database.

The difference is observable in a visual comparison of tree shapes for several randomly chosen families (Fig. 2e). The uniformity of FoldTree root-to-tip lengths appears despite the higher variance in percentage identity seen across the distribution of pairwise alignment in all families (Extended Data Figs. 1, 3 and 7). Again, when comparing to maximum-likelihood trees informed by structural characters[18], we see that adding structural information also helps creating trees with more regular root-to-tip lengths but branch lengths still have more variance than the statistically corrected pairwise distances used by FoldTree. It has been mentioned in other work[38] that high ultrametricity may be a property of using distance-based trees in general but this does not appear to be the case for LDDT-based and TM-based distance trees, which have a higher root-to-tip distance variance.

Both orthogonal metrics of adherence to a molecular clock and species tree discordance indicate that FoldTree produces trees with desirable characteristics that are ideal for constructing phylogenies with sets of highly divergent homologs. However, when considering finer grained topological differences to the species taxonomy with ASTRAL, we do observe a slightly worse performance in the OMA dataset and an equivalent performance in the CATH dataset (Extended Data Figs. 8 and 9). These results imply that FoldTree is best suited to resolving deeper evolutionary relationships and may suffer in quality compared to maximum-likelihood approaches closer to the leaves. We stress that assessing tree quality empirically without relying on an underlying model is difficult and individual phylogenies generated by any model should be interpreted using independent sources of information to verify the relative parsimony of each topology between tree inference approaches and the implied evolutionary scenario recapitulating the extant observable biology. Here, we present one such case study centered on the RRNPPA quorum-sensing systems.

## FoldTree reveals the evolutionary diversification of RRNPPA communication systems

To illustrate the potential of structural phylogenies, we reconstructed the evolutionary history of the RRNPPA family of intracellular quorum-sensing receptors in gram-positive *Bacillota* bacteria, their conjugative elements and temperate bacteriophages[25,28,32]. Recently, pioneering work combining structural comparisons among folds and sequence-based phylogenetics provided insights among some of these families[25] but a comprehensive reconstruction of the evolutionary history of this family that includes all described subfamilies[30] remains elusive.

RRNPPA receptors have a core architecture composed of five TPRs and may optionally harbor a helix–turn–helix (HTH) DNA-binding domain at the N terminus and four additional TPRs at the C terminus[25]. To avoid domain architecture changes confounding the phylogenetic signal, we applied the corecut preprocessing pipeline detailed in the Methods and Fig. 4. We then derived a FoldTree phylogeny for all of the shared core domains of the RRNPPA family. The unrooted tree delineates three main clans. Clan A is composed of only five-TPR folds (PlcR, TprA, PrgX, TraA, ComR and Rgg experimentally validated subfamilies). Clan B encompasses the nondegenerated nine-TPR folds

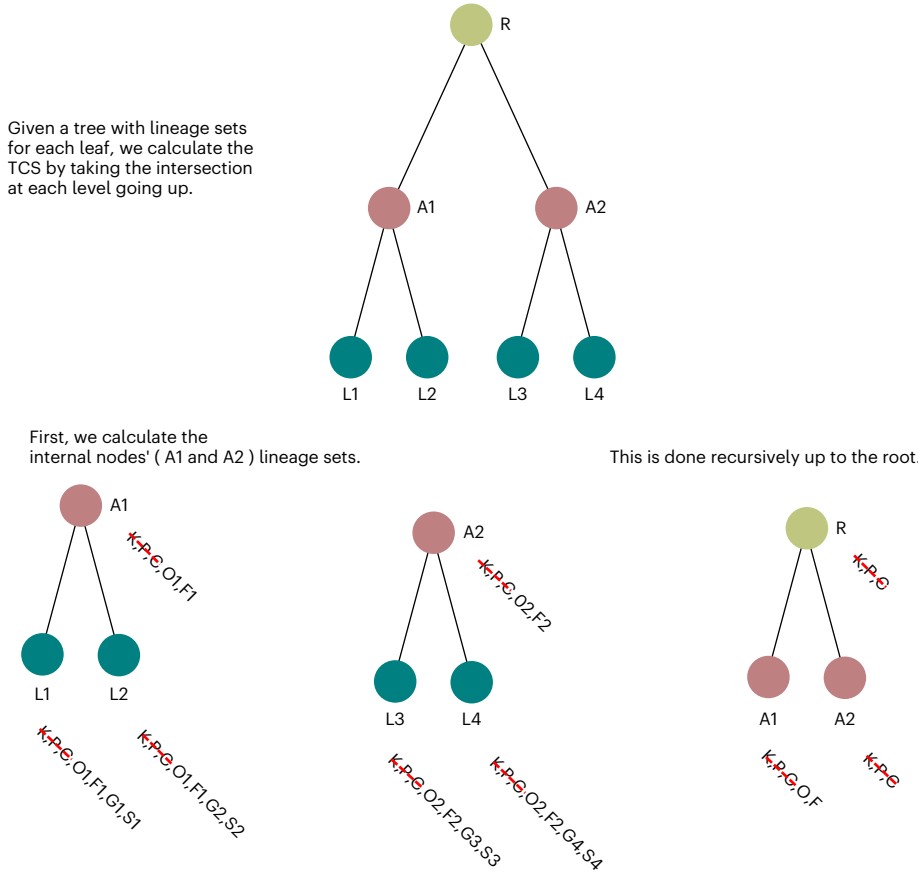

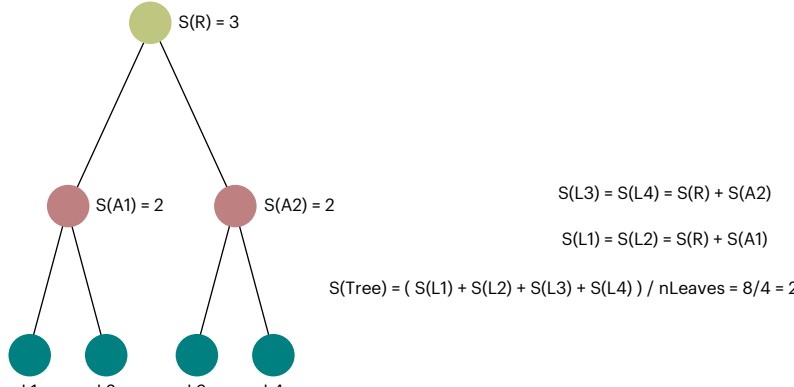

**Fig. 5 | Schema of the TCS score calculation.** In the example tree shown, each leaf is labeled with simple lineage information containing its kingdom, phylum, class, order, family, genus and species. At each level of the tree, the sets of internal nodes and the root are determined by the intersection of the two nodes found below. The leaf scores are then calculated by taking the sum of scores of internal nodes from the root to a leaf. The tree score is the sum of these scores normalized by the number of leaves. RTT, root-to-tip.

(QssR, NprR and Rap validated subfamilies). Lastly, clan C encompasses nine-TPR folds harboring more degenerated TPR motifs (AloR and AimR validated subfamilies) (Fig. 5a,d). This is more parsimonious than the scenario implied by the sequence-based tree, in which clans of five-TPR folds and nine-TPR folds are interleaved and suggest a less plausible convergent evolution of protein architectures (Extended Data Fig. 10).

According to Declerck et al.[26], receptors with degenerated TPRs correspond to a more diverged state from the last common ancestor of RRNPPA receptors than receptors with nondegenerated TPRs. In clan A of five-TPR folds, the differential degeneracy levels of TPR sequences suggest that the clade composed of PlcR and TprA receptors

less diverged from the last common ancestor of five-TPR folds than the clade composed of PrgX, TraA, ComR and Rgg receptors (Fig. 5d). While the taxonomic diversity is large in clan A (Fig. 5a,b), the PrgX–TraA–ComR–Rgg clade is specific to non-spore-forming taxa from the Lactobacillales order, suggesting indeed a recent innovation. The emergence of the PrgX–TraA–ComR–Rgg clade was accompanied by a diversification in the secretion of the communication peptide. The cognate propeptides of these receptors are either exported through the alternative PptAB translocon (as opposed to the ancestral SEC translocon)[25,28] or may correspond to leaderless communication peptides involving a yet uncharacterized secretory process[40].

The degeneracy level of TPRs is lower in receptors from clan B than in receptors from clan C, suggesting that clan B has diverged less than clan C from the last common ancestor of nine-TPR folds. Clan B encompasses substantial taxonomic diversity, with taxa distributed in both Clostridia and Bacilli classes (including many extremophiles) (Supplementary Table 1). Its topology implies that the loss of the ancestral HTH DNA-binding domain was the founding event of both the Rap subfamily and subfamily 17 of *Anoxybacter fermentans*.

Clan C contains both systems from the Paenibacillaceae taxonomic family (AloR, subfamily 08) and AimR systems used by temperate phages of Bacillaceae to regulate the lysis–lysogeny decision. Consistent with the fact that viruses evolve quickly, the AimR receptors harbor the most degenerated TPR sequences (Fig. 1d). The phylogeny reconstructed by FoldTree suggests that viral communication systems of the AimR family have been co-opted by bacteriophages from an ancestral receptor that they share with receptors of Paenibacillaceae. Consistent with this, the cognate mature communication peptides of receptors from the AimR subfamily and subfamily 08 of Paenibacillaceae are highly similar, with the widespread presence of the DPG motif across peptides of both subfamilies (Supplementary Table 1).

Whether the last common ancestor of all RRNPPA receptors had nine TPRs or five TPRs still remains an open question. While nine-TPR folds are mainly associated with dormancy regulation (for example, sporulation in bacteria and lysogeny in phages), five TPRs are mostly associated with the density-dependent regulation of biosynthetic gene clusters (Fig. 1a). Identifying an outgroup with the search of remote structural homologs could be an interesting approach to help deciphering the root and the ancestral state of the family.

## Discussion

Our exploration of the properties of scalable structural phylogenetics based on local alphabets has shown the viability of these methods on a large scale. The intrinsically slower evolutionary rate of structures allows for the reliable inference of deeper phylogenies where sequence erosion would challenge conventional sequence-based approaches. Viral evolution, quickly evolving extracellular proteins and protein families with histories stretching back to the first self-replicating cells are among the many cases that can be revisited with these techniques. As can be seen from our benchmarking efforts using thousands of families derived using structural or sequence homology, our approach remains robust at varying levels of allowable intrafamily divergence allowing for a broad applicability of structural trees all while remaining capable of resolving deeper relationships. To illustrate the utility of FoldTree, we chose to do a deep dive into one such test case, focusing on the fast-evolving RRNPPA family of cytosolic communication receptors encoded by Bacillota bacteria, their conjugative elements and their viruses. The phylogeny reconstructed by FoldTree includes all described RRNPPA subfamilies[30]. Remarkably, despite their notable divergence, the underlying diversifying history of subfamilies is more parsimonious than the sequence phylogeny in terms of taxonomy, functions and protein architectures (Extended Data Fig. 10).

Declerck et al.[26] also speculated that the level of TPR degeneracy in receptors is a marker of divergence from the last common ancestor of the family. In this respect, root-to-tip lengths are remarkably uniform throughout the entire RRNPPA structural tree with slight differences being meaningful, as the longest branches correspond to receptors with degenerated TPR sequences (Fig. 5d). Lastly, the tree inferred by FoldTree systematically is consistent with a scenario of late emergence of clades with degenerated TPRs, as a derived state of an ancestor harboring nondegenerated TPRs (Fig. 5d). Clades with degenerated TPRs have a narrow taxonomic diversity, which support the hypothesis that they are recent innovations.

Our observations regarding the RRNPPA family could only emerge once their structural homology was used to derive their evolutionary history, painting a coherent picture of their functional diversification.

We are far from the first ones to have considered protein structure in an evolutionary light. As early as 1975, Eventoff and Rossmann used the number of structurally dissimilar residues between pairs of proteins to infer phylogenetic relationships by means of a distance method[41]. This approach has been revisited to infer deep phylogenetic trees and networks using different combinations of dissimilarity measures (for example, r.m.s.d., $Q_{score}$ and $Z$ score) and inference algorithms[15,42–46]. Conformational sampling has been proposed to assess tree confidence when using this approach[14]. Some models have been developed that mathematically describe the molecular clock in structural evolution[47] or integrate sequence data with structural information to inform the likelihood of certain substitutions[48]. Other studies have modeled structural evolution as a diffusion process to infer evolutionary distances[49] or incorporated it into a joint sequence–structure model to infer multiple alignments and trees by means of Bayesian phylogenetic analysis[50,51]. Despite these efforts, to date, the quality of structure-based phylogenetics, especially compared to conventional sequence-based phylogenetics, has remained largely unknown, limiting its use to niche applications.

This work provides an extensive empirical assessment on thousands of protein families reported, using three indicators of tree quality, and demonstrates the high potential of structure-informed phylogenetics, showing the feasibility and desirability of incorporating structural information in phylogenetics. In addition, our comparison of structurally informed maximum-likelihood trees to distance-based trees shows that the simpler model has desirable performance characteristics in empirical benchmarks. The first benchmark, the TCS, measures the agreement of tree topologies with the established taxonomic classification of organisms. Individual gene trees can be expected to deviate substantially from the underlying species tree because of gene duplication, lateral transfer, incomplete lineage sorting or other phenomena. However, the evolutionary history of the underlying species will still be reflected in many parts of the tree, which is quantified by the TCS. All else being equal, tree inference approaches that tend to result in higher TCS over many protein families can be expected to be more accurate. On this metric, we obtained the best trees using FoldTree, which is based on Foldseek's structural alphabet and an alignment procedure combining structural and sequence information. After filtering lower-quality structures out of the tree-building process, tree quality improved when compared to sequence-only trees (Extended Data Fig. 3c), indicating that higher-confidence models with accurate structural information provide better phylogenetic signal. The slightly worse tree topologies of maximum-likelihood methods using partitioned structure and sequence models on the OMA benchmark compared to normal sequence-based analysis on the TCS benchmark may be because of the model's presumption of independence of the two partitions when calculating tree likelihood (when, in reality, they represent equivalent positions in the final protein). This may cause discrepancies between the partitions in likelihood calculations. When comparing this method to FoldTree using the second benchmark that also measures topological congruence with the known taxonomy, ASTRAL species tree branch support (Extended Data Figs. 8 and 9), we observe marginally worse results for FoldTree when compared to maximum likelihood on only the less divergent OMA dataset and marginally better results on the more divergent CATH dataset, indicating that FoldTree may not be the best method to deal with finer grained evolutionary histories and that maximum likelihood may offer more accurate topologies close to the leaves. However, even on the ASTRAL benchmark, 35% of tree topologies produced in the OMA dataset had better quartet support than their maximum-likelihood counterpart, indicating that there may be no 'one size fits all' solution even for protein families with homology that is readily identifiable by sequence.

The third benchmarking metric measures adherence to a molecular clock. The idea behind this metric is that the time elapsed between the first instance of a protein at the root of a tree until the

sequences observable today at the leaves should be constant for all leaves[52,53]. When considering the root-to-tip variances of the trees, the FoldTree trees adhered more closely to a molecular clock than other structural or sequence trees. We acknowledge that, in and of itself, adherence to a molecular clock is a weak indicator of the quality of a tree's topology. Nevertheless, considering the clear, consistent differences obtained, the slightly higher variance of mean percentage identity values of FoldTree alignments used as inputs for tree building (Extended Data Fig. 1) and the superiority of FoldTree when measured with the TCS criterion, the difference in adherence to a clock appears to reflect a meaningful performance increase when comparing FoldTree to the other tree inference methods.

Fold geometry evolves at a slower rate than the underlying sequence mutations[54,55]. Structural distances are, therefore, less likely to saturate over time, making it possible to align proteins correctly and recover the correct topology deeper in the tree with greater certainty. This could be observed in our results on the distant, structurally defined CATH families. Interestingly, however, FoldTree distinguished itself in TCS comparisons even at divergence times when homology is identifiable using sequence to sequence comparisons and appears only marginally worse when compared with ASTRAL. This may be because of the fact that, under conditions where the distance matrix between proteins represents a noisy version of true evolutionary distances between proteins, NJ trees are guaranteed to return the true topology below a certain level of noise[56]. We hypothesize that structural alignment rather than purely sequence-based alignment allows for the identification of truly functionally equivalent residues and that the mean rate of replacement across all amino acids of structurally equivalent positions allows for the creation of a distance matrix with lower noise that is a more accurate representation of the evolutionary time elapsed between pairs of proteins. In topologies proposed using maximum-likelihood approaches, the likelihood of a given tree is impacted by the substitutions implied by that topology across all sites. These substitutions are not considered in the context of the structure; rather, their likelihood is determined by the mean probability of all substitutions across all sites of all proteins considered when constructing the substitution rate matrix being used[57]. At larger evolutionary distances, where cumulated sequence erosion has rendered proteins unrecognizable at the sequence level, these average substitution probabilities coupled to a sequence-based alignment appear to provide a less reliable signal than a simple metric such as the percentage identity after structural alignment.

FoldTree is fine-grained enough to account for small differences between input proteins at shorter divergence times and perform in a manner comparable to state-of-the-art maximum-likelihood approaches, overcoming the often mentioned shortcoming of structural phylogenetics. However, its real strength lies in comparison at longer evolutionary distances where its topologies appear more robust than other approaches. The projection of each residue onto a structural character is locally influenced by its neighboring residues rather than global steric changes, Foldseek's representations of 3D structures are well suited to capture phylogenetic signals when comparing homologous proteins. In contrast, global structural similarity measures, such as the TM score, are confounded by conformational fluctuations involving steric changes that are much larger in magnitude than the local changes observed between functionally constrained residues during evolution. Moreover, because Foldseek represents 3D structures as strings, the computational speedups and techniques associated with string comparisons implemented in MMseqs[58] are applied to structural comparisons, making the FoldTree pipeline fast and efficient.

Recently, the fold universe was revealed using AlphaFold on the entirety of the sequences in UniProt and the ESM model[11] on the sequences in MGNIFY[59] to reach a total of nearly 1 billion structures. The UniProt structures inferred by AlphaFold were recently systematically organized into sequence-based and structure-based clusters, shedding light on undescribed fold families and their possible functions[17,60]. However useful, AI models remain imperfect representations. One improvement may be found in compensating for their limitations (for example, poor performance on membrane-bound or partially disordered regions) by omitting the structural information given by these regions. Taking this into account, in future work, it may be desirable to add an evolutionary layer of information to this exploration of the fold space using structural phylogenetics to further refine our understanding of how this extant diversity of folds emerged.

In conclusion, this work shows the potential of structural methods as a powerful tool for inferring evolutionary relationships among proteins. For relatively close proteins, structure-informed tree inference rivals sequence-only inference and the choice of approach should be tailored to the specific question at hand and the available data. For more distant proteins, structural phylogenetics opens inroads into studying evolution beyond the 'twilight' zone[61]. We believe that there remains much room for improvement in refining phylogenetic methods using the tertiary representation of proteins and hope that this work serves as a starting point for further exploration of deep phylogenies in this era of AI-generated protein structures.

## Online content

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

[1]Department of Computational Biology, University of Lausanne, Lausanne, Switzerland. [2]Swiss Institute of Bioinformatics, Lausanne, Switzerland. [3]School of Biological Sciences, Seoul National University, Seoul, South Korea. [4]Artificial Intelligence Institute, Seoul National University, Seoul, South Korea. [5]Institute of Molecular Biology and Genetics, Seoul National University, Seoul, South Korea. [6]Unidad de Bioinformática, Institut Pasteur de Montevideo, Montevideo, Uruguay. [7]Unidad de Genómica Evolutiva, Facultad de Ciencias, Universidad de la República, Montevideo, Uruguay. [8]These authors contributed equally: David Moi, Charles Bernard. ✉e-mail: dmoi@unil.ch; christophe.dessimoz@unil.ch

## Methods

No statistical methods were used to predetermine sample size.

### Pipeline description

To efficiently generate large numbers of phylogenetic trees using structure-based and sequence-based methods for comparative analysis, we built a Snakemake pipeline, which is available on the GitHub repository associated with this project (https://github.com/DessimozLab/fold_tree). The pipeline steps are outlined in Fig. 3.

The different steps of the pipeline are also usable through the Python library associated with the project, either through scripting or through the command line interface. Instructions on using the pipeline are available on the GitHub repository (https://github.com/DessimozLab/fold_tree).

**Structure trimming to 'core' region.** The corecut pipeline was used in the construction of the RRNPPA tree. This process ensures that only the domains shared among a large enough proportion of the input set of proteins are used to construct the tree. C-terminal and N-terminal domains that are outside of this consensus region are then clustered and cluster labels are used to annotate the phylogeny, showing where architecture changes may have happened. The process is schematized in Fig. 4.

Structural families often contain representatives that have extra domains not found in the majority of the members of the family. This phenomenon is especially prevalent in prokaryotes where domain-shuffling events are common. To find the conserved core of the structure set being compared and only use the phylogenetic signal from this region, we implemented a postprocessing step after the first all-versus-all comparison using Foldseek. The results of the comparisons within a set are mapped to the corresponding amino acid positions of a protein. A continuous region between the first and the last positions with over 80% of the dataset mapping to them is designated as the core. The C and N termini of the protein that are not within this core are removed and a new PDB file is generated with only the core using Biopython's structure module. The set of core proteins can then be used as input to FoldTree. The RRNPPA phylogeny was processed using this technique to reduce the input set of proteins to their core before using them as input for tree building. The trimmed regions were clustered and used to label the tree to verify that architecture changes were distributed on the tree parsimoniously.

**TCS score and root-to-tip variance calculation.** In Eq. (1) of the Supplementary Notes, we show how we calculate the TCS as a stand-in for tree quality. We also detail how root-to-tip variance is calculated in Eq. (2) of the Supplementary Notes. To provide a more intuitive way of representing how TCS is derived, we also include Fig. 2 showing a diagram of how the trees are scored recursively using sets to represent lineages.

The scores presented in the figures comparing methods are normalized by the number of leaves. The scores are impacted by the 'density' of taxonomic information in each clade rather than the size of the trees being considered. It is, therefore, only correct to consider the comparison of TCS scores within a given family with the same input set of proteins and not between families. Even if two families were to have the same number of proteins from the same species, the true topology of the tree explaining their evolution may have an intrinsically lower TCS score than another family because of the myriad evolutionary phenomena that do not follow the typical vertical mode of inheritance.

**OMA hierarchical orthologous group (HOG) selection for large-scale benchmarking.** The OMA set of protein families consists of root HOGs, which are derived from all-versus-all sequence comparisons[62]. The quest for orthologs benchmarking dataset[63] consists of 78 proteomes. The 2020 release of this dataset was used as input into the OMA orthology prediction pipeline[62] (version 2.4.1). A random selection of at most 500 random orthologous groups, chosen without user input, with at least ten proteins were compiled for each group of HOGs that were inferred to have emerged in different ancestral taxa (Bacteria, Bilateria, Chordata, Dikarya, Eukaryota, Eumetazoa, Euteleostomi, Fungi, last universal common ancestor, Opisthokonta and Tetrapoda). The UniProt identifiers of the proteins within each group were used as input to the FoldTree pipeline.

**CATH family selection for large-scale benchmarking.** CATH structural superfamilies are constructed using structural comparisons and classification[37]. Each level of classification designates a different resolution of structural similarity. These are delineated as class, architecture, topology and homology. We chose to investigate tree quality using input sets within the same homology classification and sets within the same topology. We selected a random subsample of at most 250 proteins (or the number of proteins within the family if there were fewer) from each family for 635 CATH families. Each CATH family was chosen at random without user input. The families contain the PDB identifiers and chains of the structures they correspond to. We filtered the input lists of proteins to only include one representative from each genus as defined by the taxonomic data accompanying each protein to avoid redundancy in the dataset.

The PDB files were programmatically obtained from the PDB database. The 3D structures of monomers corresponding to the chain identified in the CATH classification for each fold were extracted from PDB crystal structures using Biopython. PDBfixer from the OpenMM[64] package was used to fix crystal structures with discontinuities, nonstandard residues or missing atoms before tree building as these adversely affect structural comparisons.

**Structure tree construction.** Sets of homologous structures were downloaded from the AlphaFold Structure Database or Protein Data Bank and prepared as described above. Foldseek[17] was then used to perform an all-versus-all comparison of the structures.

Structural distances between all pairs are compiled into a distance matrix, which is used as input to QuickTree[65] to create minimum evolution trees. These trees are then rooted using the MAD method[66]. Foldseek (release: 7-04e0ec8) has two alignment modes where character-based structural alignments are performed and scored using the 3Di substitution matrix or a combination of 3Di and amino acid substitution matrices. Foldseek was used with the default weighting of amino acids and 3Di alignment during alignment scoring. A third mode, using TMalign to perform the initial alignment, was not used. It is then possible to output the fraction of identical amino acids from the 3Di and amino-acid-based alignment (Fident), the LDDT (locally derived using Foldseek's implementation) score and the TM score (normalized by alignment length). This results in a total of six structural comparison methods. We then either directly used the raw score or applied a correction to the scores to transform them to the distance matrices so that pairwise distances would be linearly proportional to time (Supplementary Methods). This resulted in a total of 12 possible Foldseek-based trees for each set of input proteins. To compile these results, Foldseek was used with alignment type 0 and alignment type 2 flags in two separate runs with the '--exhaustive-search' flag. The output was formatted to include 'lddt' and 'alntmscore' columns. The pipeline of comparing structure-based and sequence-based trees is outlined in Fig. 2.

Before starting the all-versus-all comparison of the structures, we also implemented an optional filtering step to remove poor AlphaFold models with low pLDDT values. If the user activates this option, the pipeline removes structures (and the corresponding sequences) with an average pLDDT score below 40 before establishing the final protein set and running structure and sequence tree-building pipelines. We performed similar benchmarking experiments on filtered and unfiltered versions of the OMA dataset to observe the effect of including only high-quality models in the analysis.

**Sequence-based tree construction.** Sets of sequences and their taxonomic lineage information were downloaded using the UniProt API. Clustal Omega (version 1.2.4)[67] or MUSCLE5 (version 5.0)[68] was then used to generate an MSA on default parameters. This alignment was then used with either FastTree (version 2.1)[69] on default parameters or IQ-TREE (version 1.6.12 using the flags LG + I + G) to generate a phylogenetic tree. Finally, this tree was rooted using the MAD (version 1775932) method on default parameters.

**RRNPPA phylogeny.** The metadata of 'strict' known and candidate RRNPPA quorum-sensing systems described in the RRNPP_detector paper can be found in the Supplementary Information and in the manuscript's source data[30]. The predicted regulations by quorum-sensing systems of adjacent BGCs are found in Supplementary Table 3. The propeptide sequences were downloaded from GitHub (https://github.com/TeamAIRE/RRNPP_candidate_propeptides_exploration_dataset). The 11,939 receptors listed in Supplementary Table 2 were downloaded from the National Center Biotechnology Information Genbank database and redundancy was removed by clustering at 95% identity with cd-hit[70], yielding 1,418 protein clusters. The Genbank identifiers of the 11,939 receptors were used as queries in the UniProt Retrieve/ID mapping research engine (https://www.uniprot.org/id-mapping) to retrieve corresponding UniProt or AlphaFoldDB identifiers. A total of 768 protein clusters successfully mapped to at least one UniProt or AlphaFoldDB identifier. The 768 predicted protein structures were downloaded and Foldseek was used to perform an all-versus-all comparison. On the basis of our benchmarking results, we used the Fident scores from a comparison using amino acid and 3Di alphabet alignment scoring (alignment mode 1 in Foldseek). Because this family had undergone domain architecture modifications, we decided to extract the structural region between the first and last positions of each fold where 80% of all of the other structures in the set mapped. With these core structures, we performed a second all-versus-all comparison (process outlined in Fig. 4). We again used the Fident scores (alignment mode 1) and the statistical correction from Eq. (10) in the Supplementary Notes to construct a distance matrix between the core structures. This matrix was then used with FastME[71] to create a distance-based tree. The resulting tree was annotated with ITOL[72] using the metadata available in Supplementary Table 1. To derive the sequence-based phylogeny, we built an MSA of receptors using MAFFT[73] with the parameters '--maxiterate 1000 --localpair' for high accuracy. The MSA was then trimmed with trimAl[74] under the '-automated 1' mode optimized for maximum-likelihood reconstruction. The trimmed alignment of 304 sites was given as input to IQ-TREE[2] to infer a maximum-likelihood phylogenetic under the LG + G model with 1,000 ultrafast bootstraps.

### Reporting summary

Further information on research design is available in the Nature Portfolio Reporting Summary linked to this article.

## Data availability

All UniProt identifiers necessary to replicate the experimental results are available on Zenodo (https://doi.org/10.5281/zenodo.14441021)[75]. Source data are provided with this paper.

## Code availability

The FoldTree pipeline is available on GitHub (https://github.com/DessimozLab/fold_tree). The FoldTree Jupyter notebook makes it possible to compute and visualize trees from protein structures in the browser (for example, using the Google Colab cloud service; https://colab.research.google.com/github/DessimozLab/fold_tree/blob/main/notebooks/FoldTree.ipynb). All metadata used to annotate the RRNPPA phylogeny are available in the Supplementary Data or from Zenodo (https://doi.org/10.5281/zenodo.14441021)[75].

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

## Acknowledgements

We thank the C.D. lab members for thoughtful discussions on the topic of structural evolution and their encouragement and input on this work. We also gratefully acknowledge helpful suggestions by P. Beltrao. The work was supported by Swiss National Science Foundation grants 205085 and 216623 to C.D. M.L. is a recipient of a doctoral scholarship from Agencia Nacional de Investigación e Innovación of Uruguay.

## Author contributions

D.M. designed and wrote the tree-building pipeline and analysis pipelines, collected the benchmarking data for CATH structural families, carried out the large-scale analysis for benchmarking, generated the trees for protein families, wrote the FoldTree software and drafted the manuscript. C.B. collected the data relevant to the bacterial signaling case study, analyzed and annotated the case study in light of the existing literature and wrote the corresponding sections of the paper. M.S. contributed advice and feedback on the structural distance measures evaluated in this paper and contributed to the FoldTree software. Y.N. collected HOG benchmarking data and curated examples of protein families to test the pipeline. M.L. wrote the documentation, collected the benchmarking data and curated the examples of protein families. C.D. supervised the project and

contributed to the conceptualization of the study, the interpretation of results and the manuscript writing.

## Competing interests

The authors declare no competing interests.

## Additional information

**Extended data** is available for this paper at https://doi.org/10.1038/s41594-025-01649-8.

**Correspondence and requests for materials** should be addressed to David Moi or Christophe Dessimoz.

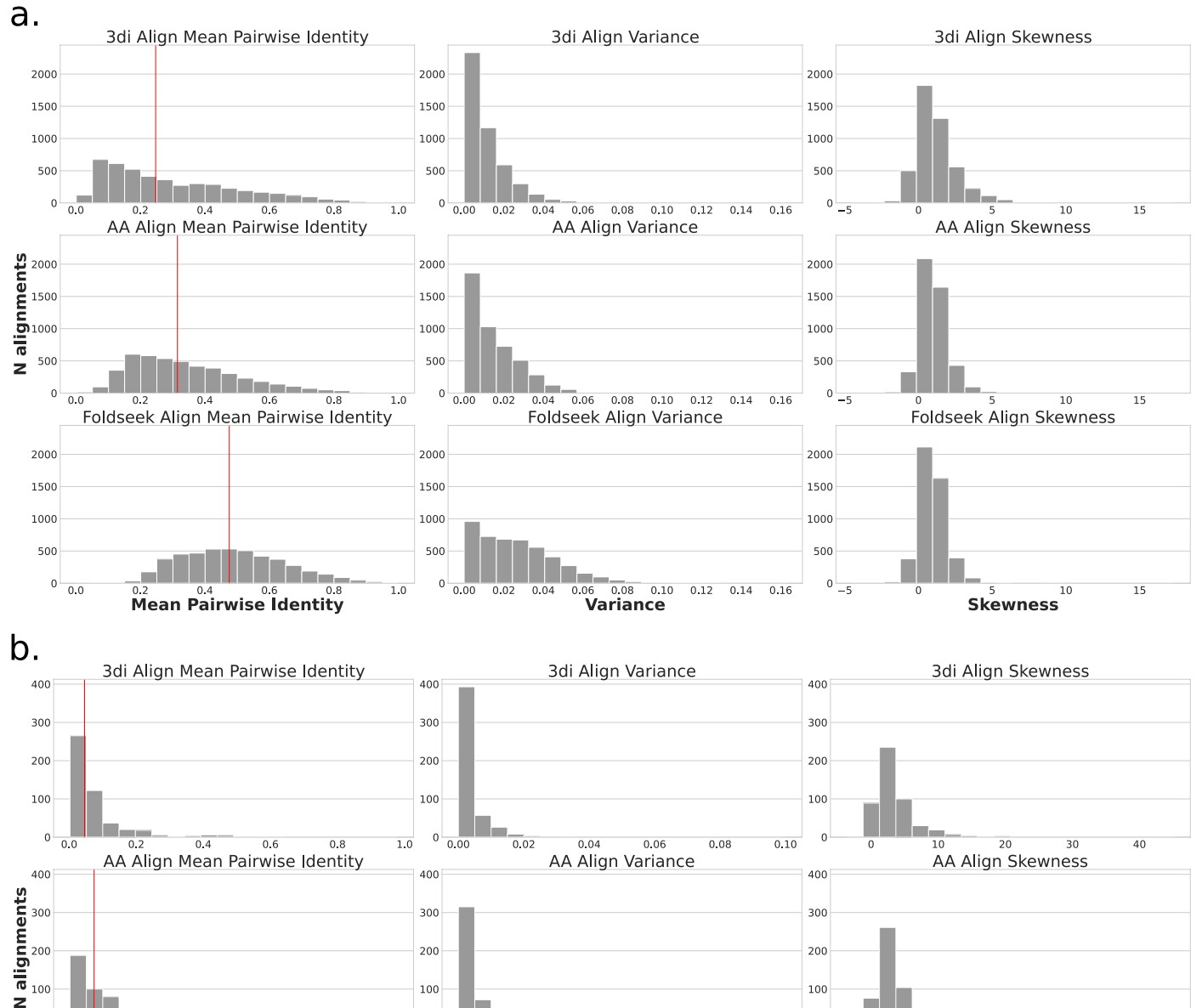

**Extended Data Fig. 1 | Alignment pairwise identity distributions aligning with amino acids, 3di or both for the CATH and OMA datasets. a.** Distributions of mean pairwise identities, variances and skewnesses are shown for all families in the OMA dataset after alignment with either 3di and amino acids (Foldseek) or with just one of the two. We observe an increase of mean pairwise identity after aligning with Foldseek accompanied by a higher variance in pairwise identities.

**b.** Aggregate distributions of the mean, variance and skewness distributions are shown for percentage identity distributions of all alignments in the CATH dataset. We observe an even larger discrepancy between the mean of pairwise identities than in a. when aligning with just one character set as opposed to both 3di and amino acid in tandem.

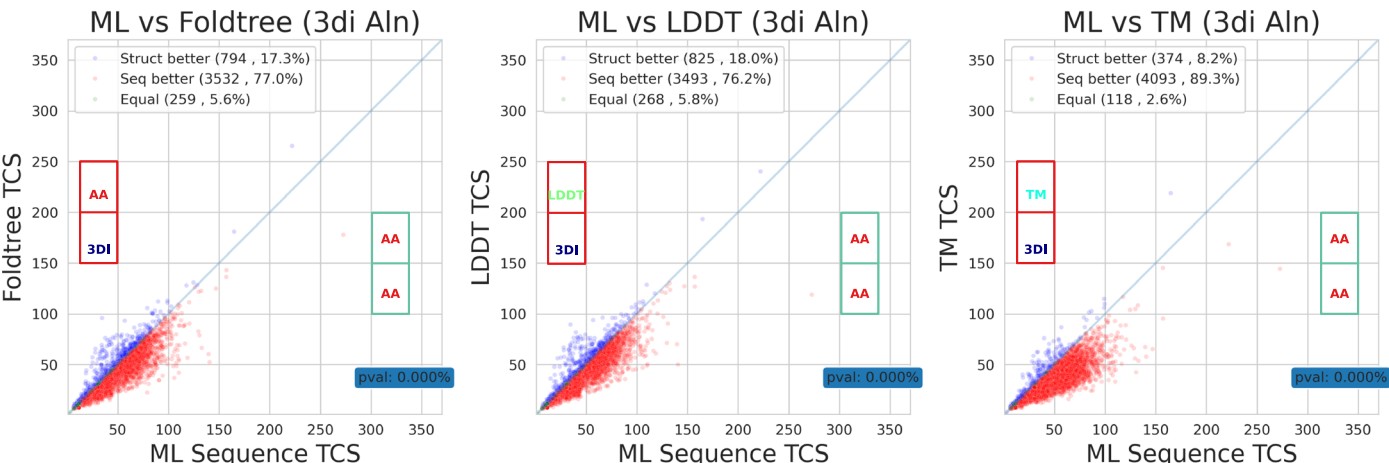

**Extended Data Fig. 2 | Structural Distance metrics using only structural character alignment score poorly in terms of TCS on the unfiltered OMA dataset.** The P-value for the Wilcoxon ranked test between the TCS scores of structure trees and their corresponding maximum likelihood trees is shown in the blue rectangle in the bottom right. With all structural distance approaches, aligning with only 3di characters produces inferior trees.

a.

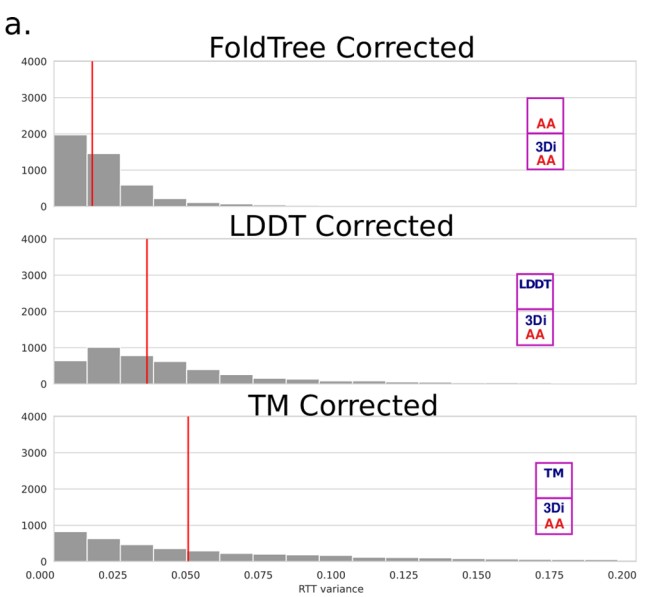
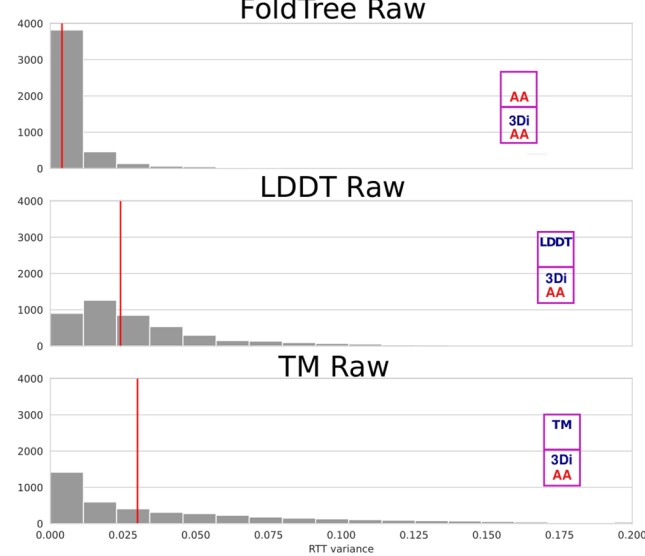

b.

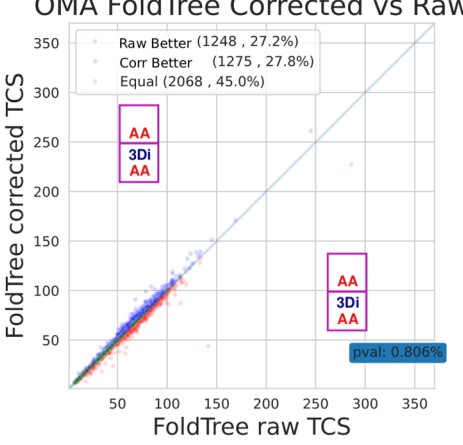
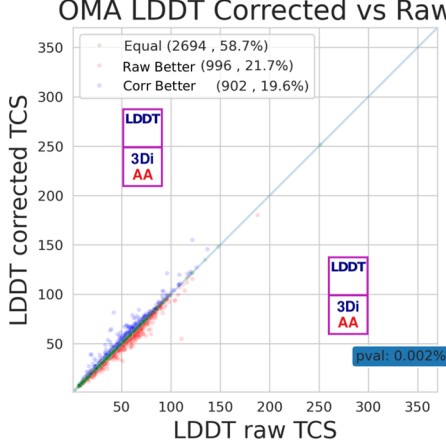
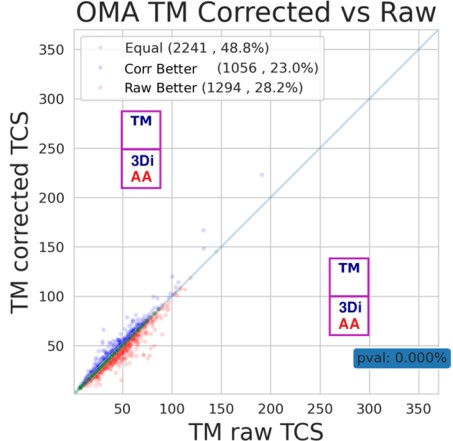

c.

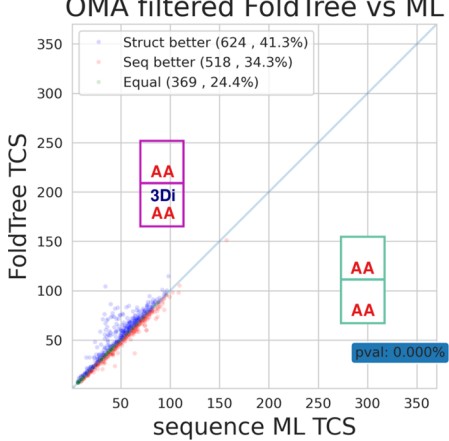
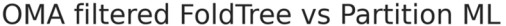
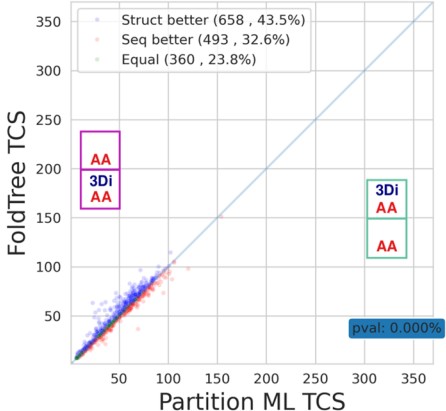

**Extended Data Fig. 3 | See next page for caption.**

**Extended Data Fig. 3 | Effect of statistical corrections on molecular clock adherence. a**. Structural distances or the fraction of identical residues in the case of FoldTree can be used to build trees without modification or statistically corrected using Supp equations 3–12 before treebuilding. Root to tip variance distributions presented in Fig. 5 are the statistically corrected versions of all distance trees. The uncorrected distances may saturate at a maximum finite distance. After statistical correction shown in equation 12 the evolutionary distance reported by FoldTree will go towards an infinite value when sequence identity approaches 7% and the statistically corrected LDDT and TM distances will approach an infinite value when measured structural similarity approaches 0. The corrected score used by FoldTree retains a lower root to tip variance than maximum likelihood approaches and the other structural distance metrics before and after correction as seen in Fig. 5. The median value of each distribution is shown with a red vertical line. **b**. The Effect of statistical corrections of structural distance metrics and the FoldTree score has a negligible impact on TCS differences in the OMA dataset. Each column of these pairwise plots shows the effect of correcting the distance metric on a TCS comparison between maximum likelihood trees and a structurally informed distance tree. The results presented in Fig. 5 are from the statistically corrected versions of the structurally informed distance trees. **c**. Performance for FoldTree and structural distance-based trees benefit from the removal of families with mean pLDDT across all proteins less than 60 and a low structural quality variance within each family (less than 10). The proportion of trees with equal or higher TCS values, when compared to standard or structurally informed maximum likelihood trees, is greater when families with poor structure quality are removed from the dataset. Misfolded regions in alphafold structures may interfere with the inference of structural characters. This in turn may negatively affect the structurally informed pairwise alignments used as input to FoldTree. The advantage appears slightly less pronounced in the case of structurally informed maximum likelihood trees.

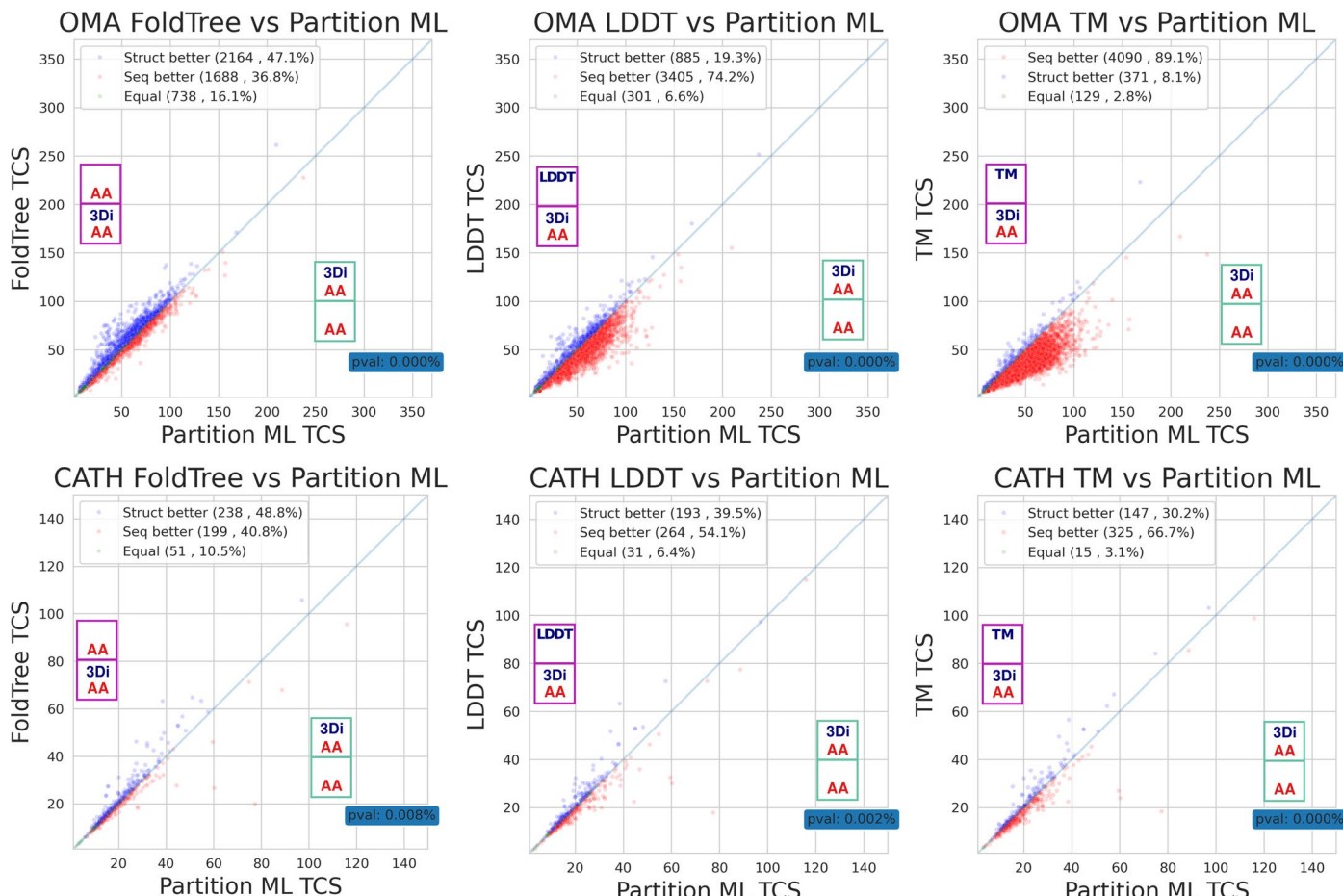

**Extended Data Fig. 4 | FoldTree and structural distance metric compared to the partition model approach presented in Puente-Lelievre et al.** FoldTree and structural distance metric-based trees are compared against a maximum likelihood model incorporating structural characters through the use of a partition model. This approach provides the evolutionary model with alignment characters which undergo a slower rate of change than amino acids. This has a positive effect on TCS values as the evolutionary divergence permitted between families increases from the OMA to the CATH dataset (Fig. 5). In a pairwise comparison between the partitioned maximum likelihood approach to geometric structural distances or the FoldTree metric, we observe that, overall, FoldTree still provides the most coherent topologies.

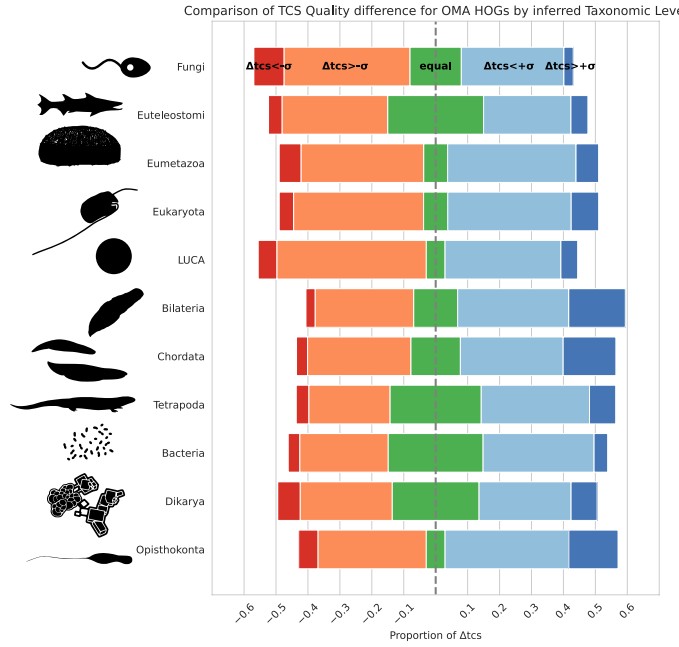

Comparison of TCS Quality difference for OMA HOGs by inferred Taxonomic Level

$$\Delta tcs(t_1, t_2) = (tcs(t_1) - tcs(t_2))/(tcs(t_1) + tcs(t_2))$$

**Extended Data Fig. 5 | TCS differences as a function of inferred emergence of OMA families.** To compare the divergence in topology quality across all families we calculated the difference in TCS scores between FoldTree and sequence-based maximum likelihood trees divided by the sum of both TCS values. Values of this score that were above one standard deviation were considered to correspond to cases where FoldTree topologies were 'much better' or 'much worse'. The input dataset of OMA families was split into subsets that were inferred to have emerged at different taxonomic levels in the unfiltered OMA dataset. The inferred time of emergence is not correlated to divergence on the protein level since some protein families are under tighter evolutionary constraints than others. Interestingly, protein families that are inferred to have emerged at the LUCA level using sequence-based homology detection appear to not have diverged greatly and do not benefit from structurally informed tree building. This may be due to the fact that they are under extreme evolutionary pressure and any modifications to this set could result in non-viability.

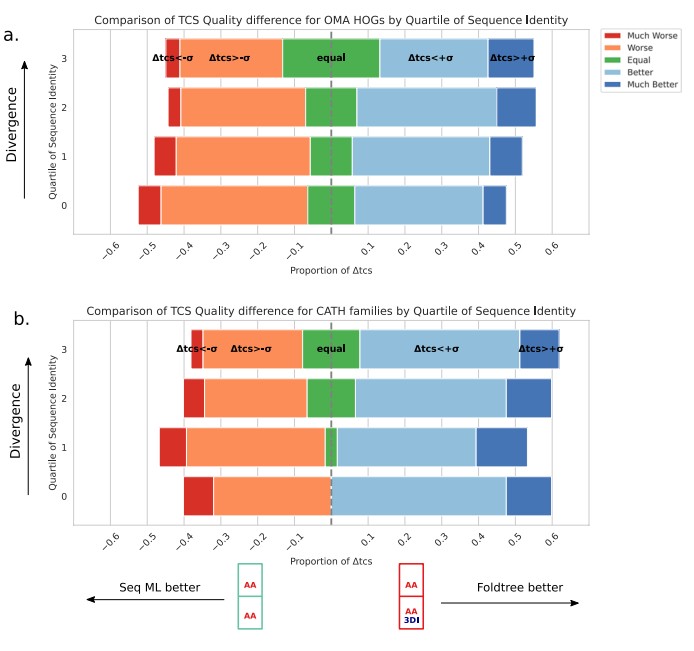

**Extended Data Fig. 6 | TCS difference as a function of alignment pairwise identity. a**) To compare the divergence in topology quality across all families we calculated the difference in TCS scores divided by the sum of both TCS values. Differences in this score that were above one standard deviation were considered to correspond to cases where FoldTree topologies were 'much better' or 'much worse' and colored in dark red or blue. We split the TCS comparisons between FoldTree and Sequence-based trees of OMA families into quartiles as a function of their mean pairwise identity after amino acid-based alignment. As alignment becomes more difficult and mean pairwise identity drops, we see the differences in tree quality between FoldTree and Sequence-based trees become more pronounced. **b**) We repeated the analysis using CATH families and found a similar trend where more divergent amino acid alignments were associated with a higher proportion of trees where FoldTree had a 'better' or 'much better' topology.

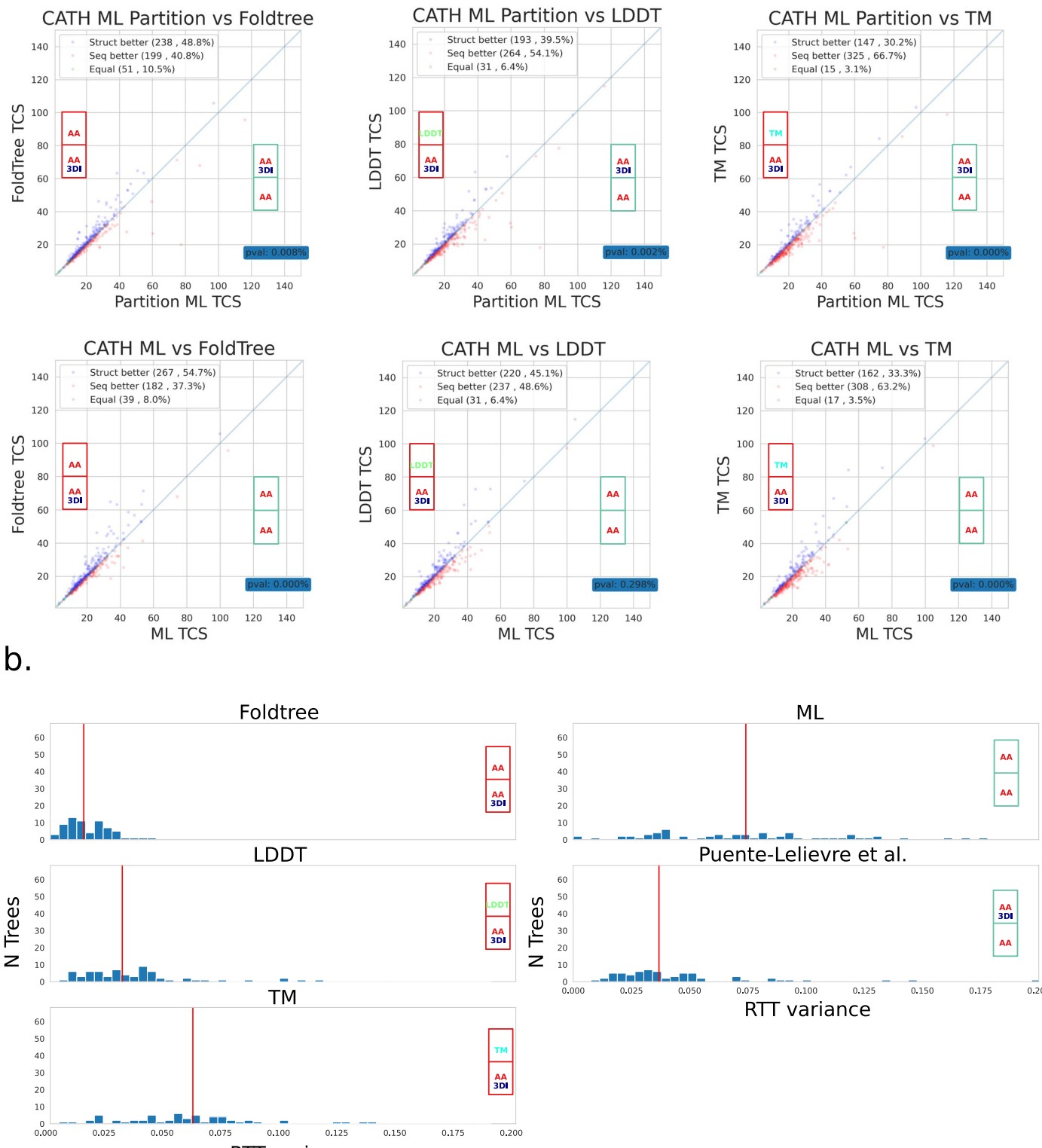

**Extended Data Fig. 7 | TCS score comparison of structural distance metrics and Sequence-based and partition model trees against FoldTree and structural distance metrics for CATH families. a**. TCS score comparison of structural distance metrics and Sequence-based and partition model trees against FoldTree and structural distance metrics for CATH families. **b**. Molecular clock adherence was quantified for the CATH using the same root to tip variance metric as in Fig. 5 (described in methods). The median value is shown with a red line. The statistically corrected FoldTree metric again shows lower root to tip distance variance when compared to the other methods.

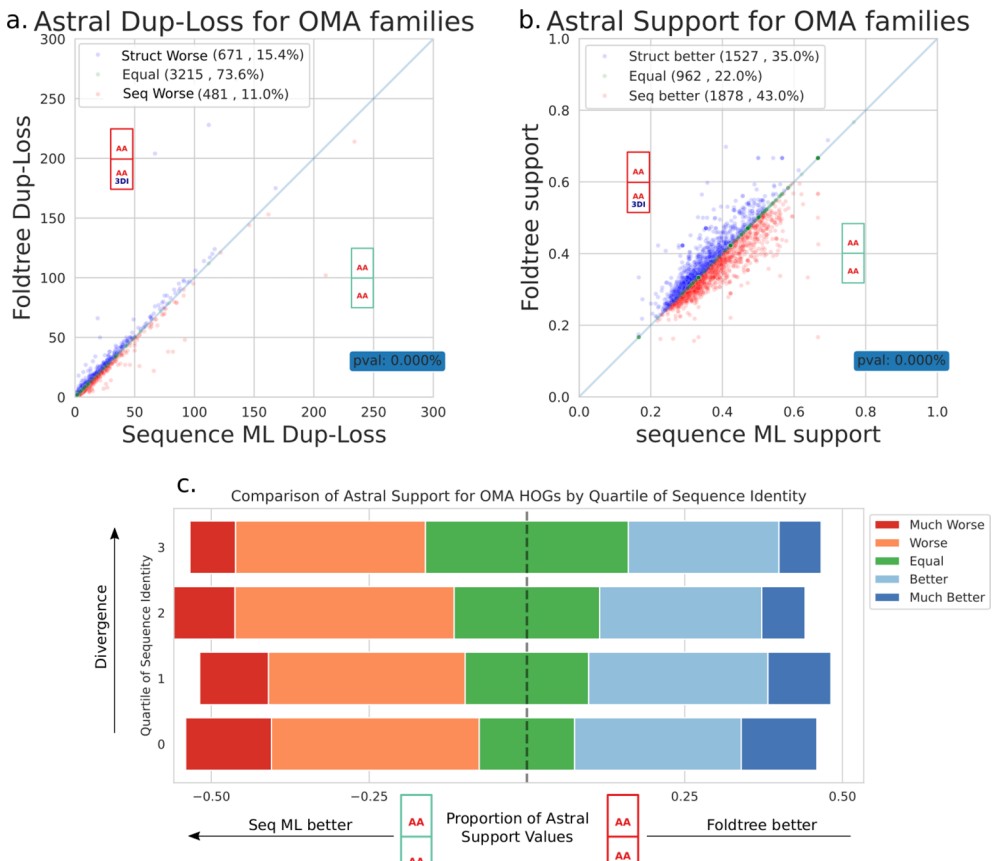

**Extended Data Fig. 8 | FoldTree topologies are compared to maximum likelihood phylogenies generated using ASTRAL quartet support for species tree branches. a.** FoldTree topologies are compared to maximum likelihood phylogenies generated using ASTRAL quartet support for species tree branches. The average support for all species tree branches is shown on the x and y axes for each family. We again observe a slight margin in favor of sequence-based trees. **b.** ASTRAL infers implied loss and duplication events in the evolutionary history of an input family by comparing a phylogeny to the species tree. Here we present the number of implied loss and duplication events for all families in the OMA dataset. A more parsimonious tree would presumably show less implied events. In this experiment only 26% of trees have a difference in terms of implied loss and duplication events with a slight margin favoring sequence-based trees. **c.** We divided the OMA dataset into quartiles of mean pairwise identity after multiple sequence alignment using amino acids. We defined the Astral comparison score as the difference of two mean support values for an input family divided by the sum of the mean support values. Any values above or below one standard deviation of this comparison score were considered 'much worse or 'much better'.

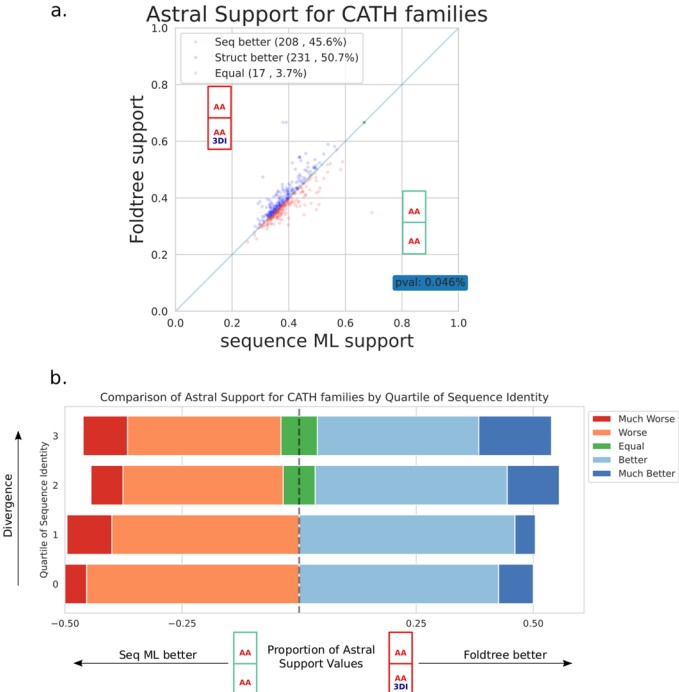

**Extended Data Fig. 9 | Difference in Mean ASTRAL support for all families in the CATH dataset. a.** Difference in Mean ASTRAL support for all families in the CATH dataset between sequence-based maximum likelihood trees and FoldTree. In contrast with the small margin in favor of sequence-based methods in the ASTRAL benchmark on all OMA families, we can observe the opposite in CATH families. **b.** We divided the CATH dataset into quartiles of mean pairwise identity after multiple sequence alignment using amino acids. As alignment pairwise identities decrease, we see the proportion of FoldTree trees with much better topologies increase. We defined the Astral comparison score as the difference of two mean branch support values for an input family divided by the sum of these mean support values. Any values above or below one standard deviation of this comparison score were considered 'much worse' or 'much better'.

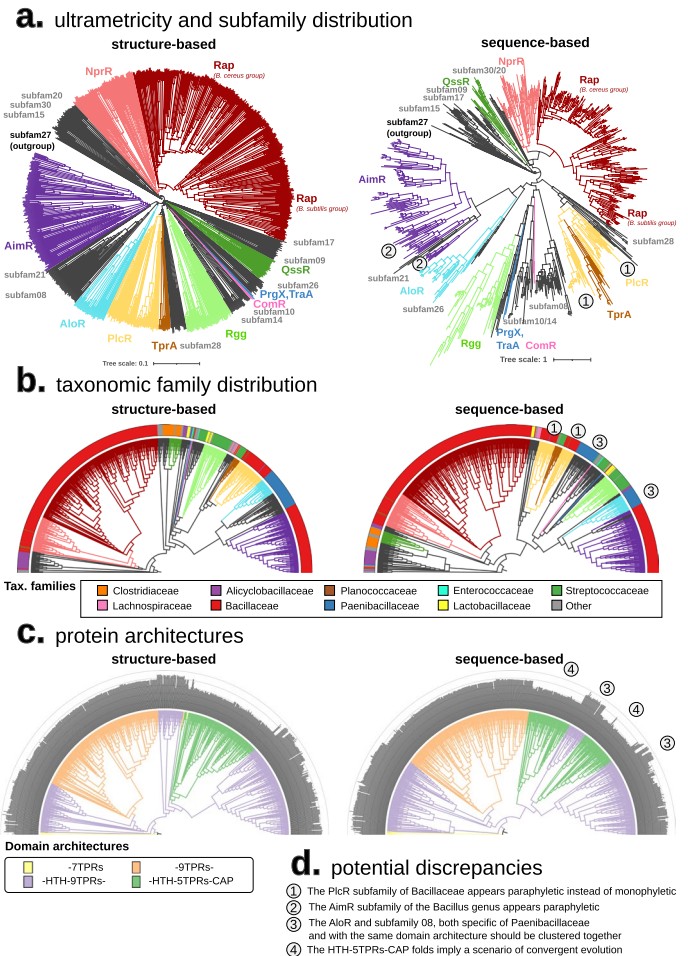

**a.** ultrametricity and subfamily distribution

**b.** taxonomic family distribution

**Tax. families**

- Clostridiaceae
- Lachnospiraceae
- Alicyclobacillaceae
- Bacillaceae
- Planococcaceae
- Paenibacillaceae
- Enterococcaceae
- Lactobacillaceae
- Streptococcaceae
- Other

**c.** protein architectures

**Domain architectures**

- -7TPRs-
- -9TPRs-
- -HTH-9TPRs-
- -HTH-5TPRs-CAP

**d.** potential discrepancies

① The PlcR subfamily of Bacillaceae appears paraphyletic instead of monophyletic
② The AimR subfamily of the Bacillus genus appears paraphyletic
③ The AloR and subfamily 08, both specific of Paenibacillaceae and with the same domain architecture should be clustered together
④ The HTH-5TPRs-CAP folds imply a scenario of convergent evolution instead of being clustered together

**Extended Data Fig. 10 | Comparison of structure- and sequence-based phylogenies of RRNPPA family. a**. Some other discrepancies can be observed on the sequence-based tree. For instance, the AimR and PlcR subfamilies are not monophyletic. Indeed, the chromosomal systems of Alkalihalobacilli from subfamily 21 branches from within the clade of AimR receptors of Bacilli temperate phages **b**. and the TprA subfamily of Streptococcaceae branches from within the PlcR clade of Bacillaceae. This is inconsistent with the literature[28,33] and not parsimonious in terms of taxonomic distribution. But most importantly, the main discrepancy of the sequence-based tree is the wrong placement of the subfamily 08 of Paenibacillaceae as a sister clade of Rgg-ComR-PrgX, with PlcR-TrpA as their outgroup (**a**). This is not parsimonious in two regards: first,

it implies that HTH-5TPRs architectures emerged two independent times **c**. and second, it implies a paraphyly of communication systems from Paenibacillaceae (**c**.). In contrast, in the structure-based tree, subfamily 08 is a sister clade of AloR, consistent with their common architecture (HTH-9TRPS) and their common specificity to the Paenibacillaceae taxonomic family. This placement, as opposed to the sequence-based tree, is more parsimonious as it implies a single emergence of the HTH-9TRPS and HTH-5TPRs folds. **d**. The discrepancies observed between the structure- and sequence-based trees are numbered sequentially, with each number mapped onto the annotated sequence-trees above for visualization.

dmoi@unil.ch

# Reporting Summary

## Statistics

For all statistical analyses, confirm that the following items are present in the figure legend, table legend, main text, or Methods section.

| n/a | Confirmed | |
|---|---|---|
| ☐ | ☒ | The exact sample size (*n*) for each experimental group/condition, given as a discrete number and unit of measurement |
| ☐ | ☒ | A statement on whether measurements were taken from distinct samples or whether the same sample was measured repeatedly |
| ☐ | ☒ | The statistical test(s) used AND whether they are one- or two-sided *Only common tests should be described solely by name; describe more complex techniques in the Methods section.* |
| ☒ | ☐ | A description of all covariates tested |
| ☒ | ☐ | A description of any assumptions or corrections, such as tests of normality and adjustment for multiple comparisons |
| ☒ | ☐ | A full description of the statistical parameters including central tendency (e.g. means) or other basic estimates (e.g. regression coefficient) AND variation (e.g. standard deviation) or associated estimates of uncertainty (e.g. confidence intervals) |
| ☒ | ☐ | For null hypothesis testing, the test statistic (e.g. *F*, *t*, *r*) with confidence intervals, effect sizes, degrees of freedom and *P* value noted *Give P values as exact values whenever suitable.* |
| ☒ | ☐ | For Bayesian analysis, information on the choice of priors and Markov chain Monte Carlo settings |
| ☒ | ☐ | For hierarchical and complex designs, identification of the appropriate level for tests and full reporting of outcomes |
| ☒ | ☐ | Estimates of effect sizes (e.g. Cohen's *d*, Pearson's *r*), indicating how they were calculated |

*Our web collection on statistics for biologists contains articles on many of the points above.*

## Software and code

Policy information about availability of computer code

| | |
|---|---|
| Data collection | Structures were downloaded from the PDB and alphafold databases. Protein families were downloaded from the CATH and OMA databases. Meta data on taxonomic lineage, sequences, protein names etc were downloaded from uniprot. |
| Data analysis | For structural search and comparison we used Foldseek. For sequence alignment we used Mafft. For tree building using maximum likelihood we used IQTREE2. For distance based trees we used quick tree. |

For manuscripts utilizing custom algorithms or software that are central to the research but not yet described in published literature, software must be made available to editors and reviewers. We strongly encourage code deposition in a community repository (e.g. GitHub). See the Nature Portfolio guidelines for submitting code & software for further information.

## Data

Policy information about availability of data

All manuscripts must include a data availability statement. This statement should provide the following information, where applicable:
- Accession codes, unique identifiers, or web links for publicly available datasets
- A description of any restrictions on data availability
- For clinical datasets or third party data, please ensure that the statement adheres to our policy

accession codes for all input data are available on zenodo. https://doi.org/10.5281/zenodo.8346286

# Research involving human participants, their data, or biological material

Policy information about studies with human participants or human data. See also policy information about sex, gender (identity/presentation), and sexual orientation and race, ethnicity and racism.

| | |
|---|---|
| Reporting on sex and gender | the study did not include human participants |
| Reporting on race, ethnicity, or other socially relevant groupings | *Please specify the socially constructed or socially relevant categorization variable(s) used in your manuscript and explain why they were used. Please note that such variables should not be used as proxies for other socially constructed/relevant variables (for example, race or ethnicity should not be used as a proxy for socioeconomic status).*<br>*Provide clear definitions of the relevant terms used, how they were provided (by the participants/respondents, the researchers, or third parties), and the method(s) used to classify people into the different categories (e.g. self-report, census or administrative data, social media data, etc.)*<br>*Please provide details about how you controlled for confounding variables in your analyses.* |
| Population characteristics | *Describe the covariate-relevant population characteristics of the human research participants (e.g. age, genotypic information, past and current diagnosis and treatment categories). If you filled out the behavioural & social sciences study design questions and have nothing to add here, write "See above."* |
| Recruitment | *Describe how participants were recruited. Outline any potential self-selection bias or other biases that may be present and how these are likely to impact results.* |
| Ethics oversight | *Identify the organization(s) that approved the study protocol.* |

Note that full information on the approval of the study protocol must also be provided in the manuscript.

# Field-specific reporting

Please select the one below that is the best fit for your research. If you are not sure, read the appropriate sections before making your selection.

☐ Life sciences ☐ Behavioural & social sciences ☒ Ecological, evolutionary & environmental sciences

For a reference copy of the document with all sections, see nature.com/documents/nr-reporting-summary-flat.pdf

# Ecological, evolutionary & environmental sciences study design

All studies must disclose on these points even when the disclosure is negative.

| | |
|---|---|
| Study description | We benchmarked the performance of phylogenetic methods which incroporated structural information and state of the art sequence-based methods in order to determine if incorporating structural information in the inference process led to better results. |
| Research sample | approximately 3000 protein families with detectable sequence homology and 500 protein families with detectable structural homology. |
| Sampling strategy | For the 3000 protein families with detectable sequence homology we used an OMA dataset for the quest for orthologs benchmark which contains a taxonomically diverse set of species. For the 500 CATH families, we sampled random families which had at least 10 genera in the family. |
| Data collection | Yannis Nevers collected the OMA data . David Moi collected the CATH data. Charles Bernard collected the RRNPPA proteins |
| Timing and spatial scale | The data was collected once at the start of the study. |
| Data exclusions | Protein families with insufficient taxonomic diversity were excluded from the CATH protein family analysis |
| Reproducibility | The identifiers of all proteins used are available on zenodo and the benchmarking pipeline code is available on github. the software environment used to produce the results is described in a yaml file which can be installed with mamba. |
| Randomization | Random families were selected from the CATH and OMA datasets. The CATH families were filtered for sufficient taxonomic breadth so as to be informative in the benchmarking experiments. |
| Blinding | We compiled results for all protein families and looked at the distributions of TCS and other metrics. |

Did the study involve field work? ☐ Yes ☒ No

# Reporting for specific materials, systems and methods

We require information from authors about some types of materials, experimental systems and methods used in many studies. Here, indicate whether each material, system or method listed is relevant to your study. If you are not sure if a list item applies to your research, read the appropriate section before selecting a response.

## Materials & experimental systems

| n/a | Involved in the study |
|-----|----------------------|
| ☒ | Antibodies |
| ☒ | Eukaryotic cell lines |
| ☒ | Palaeontology and archaeology |
| ☒ | Animals and other organisms |
| ☒ | Clinical data |
| ☒ | Dual use research of concern |
| ☒ | Plants |

## Methods

| n/a | Involved in the study |
|-----|----------------------|
| ☒ | ChIP-seq |
| ☒ | Flow cytometry |
| ☒ | MRI-based neuroimaging |

## Plants

Seed stocks | the study did not involve the use of plant samples.

Novel plant genotypes | *Describe the methods by which all novel plant genotypes were produced. This includes those generated by transgenic approaches, gene editing, chemical/radiation-based mutagenesis and hybridization. For transgenic lines, describe the transformation method, the number of independent lines analyzed and the generation upon which experiments were performed. For gene-edited lines, describe the editor used, the endogenous sequence targeted for editing, the targeting guide RNA sequence (if applicable) and how the editor was applied.*

Authentication | *Describe any authentication procedures for each seed stock used or novel genotype generated. Describe any experiments used to assess the effect of a mutation and, where applicable, how potential secondary effects (e.g. second site T-DNA insertions, mosiacism, off-target gene editing) were examined.*

