## [Peer Review File · Nature Structural & Molecular Biology]

Structural phylogenetics unravels the evolutionary diversification of communication systems in gram-positive bacteria and their viruses

Corresponding Author: Dr David Moi

Version 0:

Decision Letter:

Our ref: NSMB-A50420-T

10th Apr 2025

Dear Dr. Moi,

Thank you for submitting your revised manuscript "Structural phylogenetics unravels the evolutionary diversification of communication systems in gram-positive bacteria and their viruses" (NSMB-A50420-T). It has now been seen been re-reviewed and the comments are now below. Due to the unavailability of the original Reviewers #2 and #3 for re-review, Reviewer #1 has kindly commented on the concerns of the original Reviewer #2, and a new reviewer, Reviewer #4, has been recruited to comment on the concerns of the original Reviewer #3. The reviewers find that the paper has improved in revision, and therefore we'll be happy in principle to publish it in Nature Structural & Molecular Biology, pending minor revisions to satisfy the referees' final requests and to comply with our editorial and formatting guidelines. Notably, as per the comments of Reviewer #4, it will be critical to ensure that the Colab fold is running robustly and annotated accessibly before we can proceed with the final acceptance of the manuscript.

We are now performing detailed checks on your paper and will send you a checklist detailing our editorial and formatting requirements within the next few weeks. Please do not upload the final materials and make any revisions until you receive this additional information from us.

To facilitate our work at this stage, it is important that we have a copy of the main text as a word file. If you could please send along a word version of this file as soon as possible, we would greatly appreciate it; please make sure to copy the NSMB account (cc'ed above).

Sincerely,
Sara

Sara Osman, Ph.D.
Senior Editor
Nature Structural & Molecular Biology

Reviewer #1 (Remarks to the Author):

The authors have thoroughly addressed the comments of this and other reviewers, adding alternative phylogenetic methods and performance comparing metrics. Although I remain not quite convinced that this will become a widely used method for structural phylogenetics, it is a useful development that will stimulate further work in the field and hence merits visible publication.

I have been asked to additionally address the author's response to reviewer's #2 criticism. The main of those is that the it was not convincingly demonstrated that FoldTree, the parsimony-based approach developed by the authors, is a good approach for phylogenetic inference. This indeed was a fair criticism of the original version of the manuscript. However, in

the revision, the authors compared the performance of FoldTree to that of two more sophisticated, ML-based approaches and showed that FoldTree outperformed those. This does demonstrate that this is a reasonably robust approach even though the theoretical underpinnings of its advantages remain unclear. I nevertheless believe that the criticisms have been adequately addressed. The authors admit that their methodology is far from being the last word in structural phylogenetics. Perhaps, this has to be made even clearer in the manuscript itself.

Reviewer #4 (Remarks to the Author):

This is a very interesting manuscript. I came in later as a reviewer on the structural aspect of the work and hope I will be able to add towards the assessment from a structural perspective. Therefore, you will find me both looking at the manuscript and also reviewer #3's comments.

B. Originality and significance:

Thank you for adding more references – what I am missing is your counter argument. I get both the argument, that is concept has been proposed before, as also your argument that these kind of analyses just now becomes availability. Can the authors sharpen the argument what is new through their analysis? If the protein alphabet used is the key to this innovation – that's what I would like to see re-sharpened.

C. Data and methodology

In agreement with reviewer #3 I find some colloquial comments that I don't find informative when reading. Examples are: Page 5 third paragraph "a battery of methods" – a descriptive number, or naming the methods I would have found for precise, or page 7 "FoldTree has the lowest root-to-tip variance" – but is it significant and meaningful?

On the comment of structural similarity may not be informative – why not make this argument in the manuscript? There is no harm done by detailing a possible limitation.

Overall, I think some of the arguments of reviewer #3 could be discussed more detailed in the manuscript.

On a general note, I have the question to the authors: AlphaFold by itself has biases, such as the training base. As the method is structure-based – what will the authors expect in which way this biases the prediction?

Reviewer #4 (Remarks on code availability):

I ran the Colab fold – for the example enters it generates tree. That's good. It still throws a lot of messages and some errors in step3 – for a non-expert user it would be great to annotate the errors more detailed. When I changed the Uniprot ID the whole colab fold ran into an error and could not obtain a tree. For non-expert user usage this needs reworking and annotating of errors.

Version 1:

Decision Letter:

7th Jul 2025

Dear Dr. Moi,

We are now happy to accept your revised paper "Structural phylogenetics unravels the evolutionary diversification of communication systems in gram-positive bacteria and their viruses" for publication as a Article in Nature Structural & Molecular Biology.

To assist our authors in disseminating their research to the broader community, our SharedIt initiative provides all co-authors

with the ability to generate a unique shareable link that will allow anyone (with or without a subscription) to read the published article. Recipients of the link with a subscription will also be able to download and print the PDF.

Your paper will be published online soon after we receive proof corrections and will appear in print in the next available issue. You can find out your date of online publication by contacting the production team shortly after sending your proof corrections.

Sincerely,
Sara

Sara Osman, Ph.D.
Senior Editor
Nature Structural & Molecular Biology

Reviewer #1:

Remarks to the Author:

The authors have thoroughly addressed the comments of this and other reviewers, adding alternative phylogenetic methods and performance comparing metrics. Although I remain not quite convinced that this will become a widely used method for structural phylogenetics, it is a useful development that will stimulate further work in the field and hence merits visible publication.

I have been asked to additionally address the author's response to reviewer's #2 criticism. The main of those is that the it was not convincingly demonstrated that FoldTree, the parsimony-based approach developed by the authors, is a good approach for phylogenetic inference. This indeed was a fair criticism of the original version of the manuscript. However, in the revision, the authors compared the performance of FoldTree to that of two more sophisticated, ML-based approaches and showed that FoldTree outperformed those. This does demonstrate that this is a reasonably robust approach even though the theoretical underpinnings of its advantages remain unclear. I nevertheless believe that the criticisms have been adequately addressed. **The authors admit that their methodology is far from being the last word in structural phylogenetics. Perhaps, this has to be made even clearer in the manuscript itself.**

We thank reviewer #1 for their positive appraisal of our efforts to complete a thorough benchmark. We have clarified the fact that this approach is only a first attempt in the discussion section of the manuscript.

In conclusion, this work shows the potential of structural methods as a powerful tool for inferring evolutionary relationships among proteins. For relatively close proteins, structure-informed tree inference rivals sequence-only inference, and the choice of approach should be tailored to the specific question at hand and the available data. For more distant proteins, structural phylogenetics opens new inroads into studying evolution beyond the “twilight” zone(Rost, 1999). We believe that there remains much room for improvement in refining phylogenetic methods using the tertiary representation of proteins and hope that this work serves as a starting point for further exploration of deep phylogenies in this new era of AI-generated protein structures.

Reviewer #2:

None

Reviewer #3:

None

Reviewer #4:

Remarks to the Author:

This is a very interesting manuscript. I came in later as a reviewer on the structural aspect of the work and hope I will be able to add towards the assessment from a structural perspective. Therefore, you will find me both looking at the manuscript and also reviewer #3's comments.

B. Originality and significance:

Thank you for adding more references – what I am missing is your counter argument. I get both the argument, that is concept has been proposed before, as also your argument that these kind of analyses just now becomes availability. **Can the authors sharpen the argument what is new through their analysis?**

We have emphasized that the novelty of the current work is the large scale benchmarking of methods which indicates that structure is providing trees with measurably better topologies and ultrametricity. This is elaborated on in the discussion section. We explain how this large scale effort lays a foundation and benchmarking strategy in the absence of an explicit structural evolutionary model.

Here, we report the large-scale comprehensive evaluation of phylogenetic trees reconstructed from the structures of thousands of protein families across the tree of life, using multiple kinds of distance measures and tree building strategies. We tested nine structure-informed approaches, using divergence measures obtained using Foldseek(van Kempen et al., 2023), which outputs scores from rigid body alignment, local superposition-free alignment and structural alphabet-based sequence alignments. In addition, we tested a recently proposed partitioned structure and sequence likelihood method(Puente-Lelievre et al., 2023). The performance of these approaches has been previously assessed on the task of detecting whether folds are homologous and belong to the same family(Mariani et al., 2013; van Kempen et al., 2023; Xu & Zhang, 2010), or on a few examples(Puente-Lelievre et al., 2023), but have never been systematically evaluated for phylogenetic tree inference. Remarkably, we found that some, though not all, structure-informed approaches are competitive with state-of-the-art sequence based phylogenetic methods, and outperform them on highly divergent datasets across benchmarks related to tree topology (TCS and ASTRAL-based species tree branch support) as well as testing the adherence to a molecular clock.

If the protein alphabet used is the key to this innovation – that's what I would like to see re-sharpened.

C. Data and methodology

In agreement with reviewer #3 I **find some colloquial comments that I don't find informative when reading. Examples are: Page 5 third paragraph "a battery of methods" – a descriptive number, or naming the methods I would have found for precise,**

We now explicitly mention the number of methods reported in the manuscript:

For trees reconstructed from closely related protein families using standard sequence alignments (the 'OMA dataset', see methods), we thoroughly tested nine approaches using combinations of structure and sequence input data paired with either multiple sequence alignment (MSA) and maximum likelihood (ML) approaches to build trees or distance based neighbor joining tree building (NJ) (Figure 1a and 3).

or page 7 “FoldTree has the lowest root-to-tip variance” – but is it significant and meaningful?

We have added some comments on this feature of the trees as well as a reference.

Again, when comparing to maximum likelihood trees informed by structural characters (Puente-Lelievre et al., 2023), we see that adding structural information also helps creating trees with more regular root to tip lengths but branch lengths still have more variance than the statistically corrected pairwise distances used by FoldTree. It has been mentioned in other work (Mutti et al., 2024b) that high ultrametricity may be a property of using distance-based trees in general, but this does not appear to be the case for LDDT and TM-based distance trees which have a higher root to tip distance variance.

On the comment of structural similarity may not be informative – why not make this argument in the manuscript? **There is no harm done by detailing a possible limitation.** Overall, I think some of the arguments of reviewer #3 could be discussed more detailed in the manuscript.

On a general note, I have the question to the authors: AlphaFold by itself has biases, such as the training base. As the method is structure-based – what will the authors expect in which way this biases the prediction?

Comments regarding this limitation have been made in the discussion section:

However useful, AI models remain imperfect representations. One improvement may be found in compensating for their limitations (e.g. poor performance on membrane bound or partially disordered regions), by omitting the structural information given by these regions. Taking this into account, in future work it may be desirable to add an evolutionary layer of information to this exploration of the fold space using structural phylogenetics to further refine our understanding of how this extant diversity of folds emerged.

Remarks on code availability:

I ran the Colab fold – for the example enters it generates tree. That's good. It still throws a lot of messages and some errors in step3 – for a non-expert user it would be great to annotate the errors more

detailed. When I changed the Uniprot ID the whole colab fold ran into an error and could not obtain a tree. **For non-expert user usage this needs reworking and annotating of errors.**

Using random uniprot IDs will not work. The colab now has additional instructions regarding the use of AFDB cluster IDs. These cluster IDs are the uniprot ID of the cluster representative stored in the AFDB cluster database. Please use either a list of uniprot IDs or an AFDB cluster ID. A single uniprot ID is not a correct input to the method.

Input Types

This notebook supports three types of input for constructing phylogenetic trees from protein structures:

- **AFDB Cluster**: Use this option to input an AlphaFold Database (AFDB) cluster. You must provide a valid AFDB cluster ID (e.g., `A0A074YNE0`). Only AFDB cluster IDs are accepted for this input type.
- **Identifier List**: Provide a list of UniProt IDs, one per line. It is the user's responsibility to ensure that the listed proteins are homologous.
- **Custom PDBs**: Upload your own set of PDB files. All PDB files must be compressed into a single `.zip` archive before uploading. As with the identifier list, users must verify that the input proteins are homologous.

Carefully select the input type that matches your data and ensure the biological relevance of your input set.